psychology/behaviour

personality, neuroticism, extraversion, noise sensitivity, noise tolerance, speech-in-noise comprehension

**Author for correspondence:**
Malte Wöstmann
e-mail: malte.woestmann@uni-luebeck.de

# Personality captures dissociations of subjective versus objective hearing in noise

Malte Wöstmann[1,2], Julia Erb[1,2], Jens Kreitewolf[1,2,3,4] and Jonas Obleser[1,2]

[1]Department of Psychology, and [2]Center of Brain, Behavior, and Metabolism, University of Lübeck, Lübeck, Germany
[3]Department of Psychology, and [4]Department of Mathematics and Statistics, McGill University, Montreal, Quebec, Canada

MW, 0000-0001-8612-8205; JE, 0000-0002-3440-7269;
JK, 0000-0002-8569-1554; JO, 0000-0002-7619-0459

Acoustic noise is pervasive in human environments. Some individuals are more tolerant to noise than others. We demonstrate the explanatory potential of Big-5 personality traits neuroticism (being emotionally unstable) and extraversion (being enthusiastic, outgoing) on subjective self-report and objective psycho-acoustic metrics of hearing in noise in two samples (total $N = 1103$). Under statistical control for demographics and in agreement with pre-registered hypotheses, lower neuroticism and higher extraversion independently explained superior self-reported noise resistance, speech-hearing ability and acceptable background noise levels. Surprisingly, objective speech-in-noise recognition instead increased with higher levels of neuroticism. In turn, the bias in subjectively overrating one's own hearing in noise decreases with higher neuroticism but increases with higher extraversion. Of benefit to currently underspecified frameworks of hearing in noise and tailored audiological treatments, these results show that personality explains inter-individual differences in coping with acoustic noise, which is a ubiquitous source of distraction and a health hazard.

## 1. Introduction

Exposure to acoustic noise causes damage to the auditory system but is also associated with non-auditory health issues (for review, see [1]), such as cardiovascular disease [2] or sleep disturbance [3]. People who report higher noise sensitivity are more vulnerable to the adverse effects of noise [4]. Noise sensitivity has been isolated

as a stable trait and different methods have been developed to assess it, primarily using self-report [5,6]. It is unclear how public health and audiological rehabilitation should take into account the considerable inter-individual differences in noise sensitivity. Most importantly, any such considerations need a solid framework based on the predictors of differences in noise sensitivity [7–9].

Noise sensitivity explains inter-individual differences in listening behaviour, such that higher noise sensitivity relates to e.g. increased use of hearing protection [10], increased need for privacy when receiving a call [11] and decreased time spent listening passively to music [12]. Therefore, hearing in noise is an essential factor in understanding human psychological experience and behavioural reactions in ubiquitous noise-contaminated environments.

It has long been known [5] and is well-established that personality is a powerful predictor of some particular self-report measures of hearing in noise (for review, see [13]). Compromised self-reported hearing in noise was found to correlate with higher trait neuroticism (i.e. being anxious; worrying; emotionally less stable) and lower trait extraversion (i.e. being less enthusiastic; less outgoing; e.g. [14–16]). But does the explanatory power of personality hold for audiological screening tests of hearing in noise and, most importantly, for objective measures of hearing in noise? To address these questions with robust effect size estimates, we use a large sample (total $N = 1103$) and statistical control of potential confounders (such as age, gender, education and other personality dimensions).

In the field of audiology, it has proven valid to differentiate subjective (self-rated) from objective (performance-based) hearing ability (e.g. [17,18]), which diverge considerably as a function of demographics and other non-auditory factors [19,20]. Regarding subjective and objective hearing in noise, research has established some association between the two, such that listeners with poorer subjective hearing in noise perform worse on certain cognitive tasks in the presence of acoustic noise [21]. In general, however, subjective hearing in noise weakly relates to objective measures, such as sensory measures of auditory processing [22] or the degree of memory disruption by task-irrelevant speech [23]. A prevalent view is that subjective hearing in noise relates primarily to evaluative (rather than sensory) aspects of auditory processing.

Understanding the relation between subjective and objective auditory processing and personality traits is crucial for hearing-rehabilitation healthcare [24]. Different factors such as awareness (on the sides of both patients and clinicians), access or expectations hamper effective treatment of hearing loss [25]. If treated, a common problem is poor compliance with hearing aids [26,27]. Also, for individuals who experience limited benefit from hearing aids for speech perception in noise, this limitation is poorly explained by audiological or demographic factors alone, underlining the importance of examining non-audiological influences [28,29]. Understanding the contribution of personality to hearing in noise may aid the clinical management of hearing loss and open novel routes to more suitable, individualized audiological treatment.

The goal of the present study was to establish robust estimates of the association of personality traits neuroticism and extraversion with recognized subjective and objective measures of hearing in noise in a large sample, under statistical control for demographic factors and other personality traits. In agreement with pre-registered hypotheses (osf.io/fgyj9), lower neuroticism and higher extraversion were associated with better subjective hearing in noise. To our surprise, these relationships reversed for objective hearing in noise. In other words, we found underrated (i.e. lower subjective than objective) hearing in noise with higher neuroticism and overrated hearing in noise with higher extraversion. We focused particularly on these two personality dimensions since both have been consistently related to subjective hearing in noise in the literature (but also included remaining Big-5 dimensions agreeableness, openness and conscientiousness).

## 2. Results

We conducted an online study to assess the explanatory power of Big-5 personality traits (neuroticism, extraversion, openness, agreeableness, conscientiousness) for established, subjective and objective measures of hearing in noise in a sample of $N = 1103$ participants (figure 1a). We tested the hypotheses that lower neuroticism and higher extraversion are both associated with better subjective and objective hearing in noise.

### 2.1. Validation of personality and hearing in noise tests

In agreement with previous research [30], a short version of the Big-5 personality inventory exhibited satisfying levels of internal consistency (CA: Cronbach's alpha; $CA_{Neuroticism} = 0.73$, $CA_{Extraversion} = 0.8$,

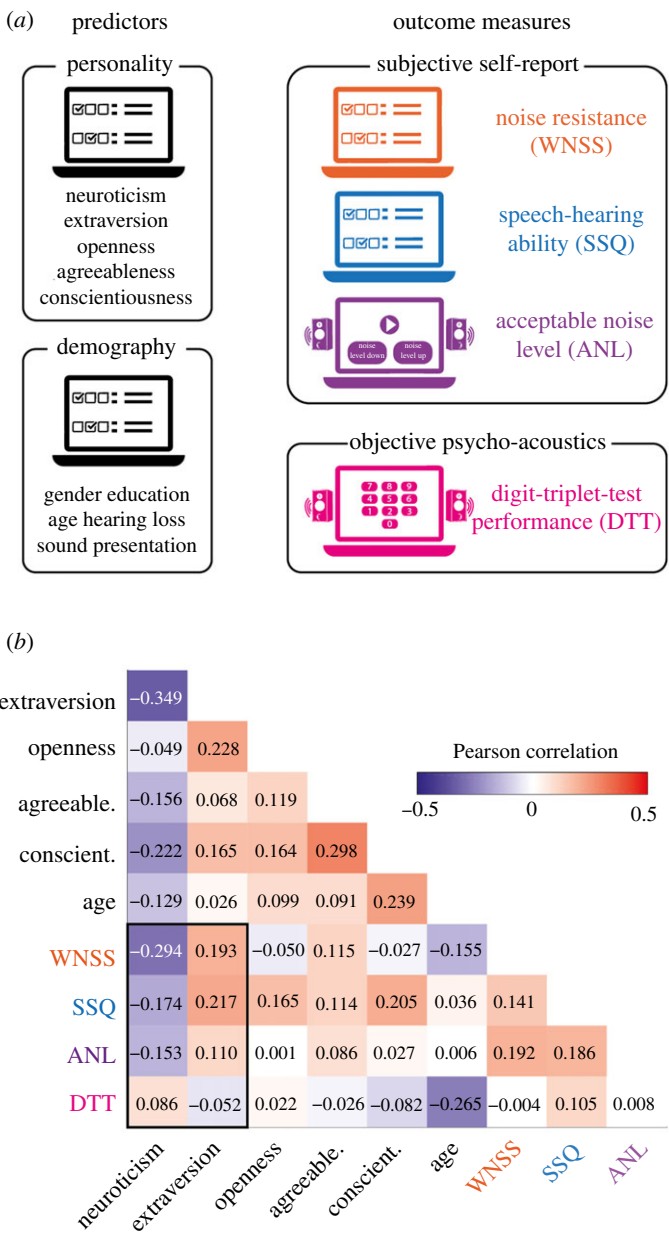

**Figure 1.** Study design and bivariate correlations. (*a*) The present online study was designed to use the predictors personality (Big-5 dimensions) and demographic information to model results of four established hearing in noise tests. Subjective reports of hearing in noise probed noise resistance using scores of the Weinstein noise sensitivity scale (WNSS; inverted), speech-hearing ability using scores from the speech spatial and qualities scale (SSQ, Part 1: Speech), and the acceptable noise level (ANL), which corresponds to the signal-to-noise ratio (SNR) (inverted) of speech in multi-talker background noise. Objective hearing in noise was assessed with the digit-triplet-test (DTT), which determines the SNR (inverted) required to correctly report 50% of digit triplets presented in noise (speech reception threshold, $SRT_{50}$). (*b*) Correlation matrix ($N = 1103$ participants), showing Pearson correlation coefficients. Categorical variables (gender, education, hearing loss and sound presentation) are not shown. The black box highlights the correlations of major interest of personality dimensions neuroticism and extraversion with hearing in noise tests. Electronic supplementary material, figure S1 shows histograms of all variables in *b* and scatterplots for all bivariate relations. Electronic supplementary material, figure S2 shows the relation of neuroticism with DTT performance for individual trials of the DTT.

$CA_{Openness} = 0.67$, $CA_{Agreeableness} = 0.56$, $CA_{Conscientiousness} = 0.6$). As common and as expected, the Big-5 personality dimensions showed small-to-moderate pairwise correlations (figure 1*b*; all $|r| < 0.35$) but were sufficiently independent to investigate their individual associations with noise tolerance.

Separate facets of hearing in noise were measured using four tests. Subjective self-report measures of hearing in noise were assessed using (i) noise resistance on the Weinstein noise sensitivity scale (WNSS, inverted scores; $CA_{WNSS} = 0.88$), (ii) speech-hearing ability using the speech, spatial and qualities of hearing scale (SSQ, Part 1: Speech; $CA_{SSQ} = 0.79$) and (iii) the self-adjusted, maximally acceptable noise

level (ANL) while listening to speech. Additionally, a psycho-acoustic test quantified objective hearing in noise as (iv) the noise level at which listeners correctly reported 50% of presented digit triplets (DTT, inverted scores; test–retest reliability across two runs: $r = 0.52$). Small pairwise correlations between these tests (figure 1b; all $|r| < 0.2$) confirmed that they assess partly independent facets of hearing in noise.

## 2.2. Planned analyses: relation of personality and hearing in noise

In agreement with our hypotheses and with previous evidence from smaller samples (e.g. [14–16]), bivariate correlations showed that higher self-reported noise resistance (WNSS) was associated with lower neuroticism ($r = -0.294$; $p < 0.001$; CL = 0.405) but with higher extraversion ($r = 0.193$; $p < 0.001$; CL = 0.562). To estimate the size of these effects, we report the common language (CL) effect size, which here indicates the following: for a randomly sampled person with above-average self-reported noise resistance (WNSS), the probability for an above-average neuroticism score is about 41%, but about 56% for an above-average extraversion score.

Beyond previous research, we found that better self-reported speech-hearing ability (SSQ) and higher self-adjusted acceptable levels of background noise (ANL) were related to lower neuroticism (SSQ: $r = -0.174$; $p < 0.001$; CL = 0.444; ANL: $r = -0.153$; $p < 0.001$; CL = 0.451) and to higher extraversion (SSQ: $r = 0.217$; $p < 0.001$; CL = 0.57; ANL: $r = 0.110$; $p = 0.003$; CL = 0.535). Interestingly, and against what we had hypothesized, this pattern of results showed a tendency to reverse for objective hearing in noise in the digit-triplet-test (DTT): Correct report of digit triplets at higher background noise levels was associated with higher neuroticism ($r = 0.086$; $p = 0.006$; CL = 0.527). However, extraversion was not significantly associated with DTT performance ($r = -0.052$; $p = 0.101$; CL = 0.484).

To identify independent contributions of personality dimensions in explaining variance in hearing in noise, we performed four multiple regression analyses to explain scores of four hearing in noise tests by personality dimensions, controlling for potential confounders age, gender, education (highest school-leaving qualification), sound presentation (speakers versus headphones) and self-reported hearing loss (figure 2 and table 1). Goodness-of-fit of regression models indicated that together, the predictors explained about 6–16% of variance in hearing in noise ($R^2_{\text{WNSS}} = 0.164$; $R^2_{\text{SSQ}} = 0.139$; $R^2_{\text{ANL}} = 0.058$; $R^2_{\text{DTT}} = 0.143$).

Overall, results of multiple regression analyses largely converged with bivariate correlations: under statistical control for other predictors, neuroticism negatively predicted noise resistance (WNSS; $\beta = -0.279$; $p < 0.001$; CL = 0.416), speech-hearing ability (SSQ; $\beta = -0.107$; $p = 0.001$; CL = 0.467) and the acceptable noise level (ANL; $\beta = -0.114$; $p = 0.005$; CL = 0.467), but positively predicted objective performance of speech-in-noise recognition (DTT; $\beta = 0.066$; $p = 0.045$; CL = 0.52). Conversely, extraversion positively predicted noise resistance (WNSS; $\beta = 0.118$; $p < 0.001$; CL = 0.537), speech-hearing ability (SSQ; $\beta = 0.127$; $p < 0.001$; CL = 0.539) and the acceptable noise level (ANL; $\beta = 0.087$; $p = 0.034$; CL = 0.525), but was not significantly related to objective hearing in noise in the digit-triplet-test (DTT; $\beta = -0.028$; $p = 0.399$; CL = 0.491).

Following our pre-registered analysis plan (osf.io/fgyj9), we estimated the replicability of obtained results by division of the total sample into the two cohorts of participants ($N_{\text{cohort 1}} = 554$; $N_{\text{cohort 2}} = 549$). Figure 2 shows overall good convergence of the explanatory power for individual predictors across the two cohorts. It might be surprising at first glance that the modelled relation of neuroticism and objective hearing in noise (DTT) was clearly positive in the first cohort but much reduced in the second cohort (figure 2a, top right). However, random sub-sampling from the total sample (10 000 sub-samples of $N_{\text{sub-sample}} = 550$) revealed that this relation was somewhat overestimated in cohort one (only 3.28% of sub-samples resulted in a $\beta_{\text{sub-sample}} \geq \beta_{\text{cohort 1}}$) and underestimated in cohort two (only 2.43% of sub-samples resulted in a $\beta_{\text{sub-sample}} \leq \beta_{\text{cohort 2}}$). Most importantly, across all sub-samples, the relation of neuroticism and objective hearing in noise (DTT) was significantly positive (mean $\beta_{\text{sub-sample}} = 0.066$; s.d. = 0.029; 95% CI = [0.011, 0.123]).

## 2.3. Unplanned analyses: dissociations of subjective versus objective hearing in noise

The data as analysed thus far reveal some dissociations in magnitude and direction with which personality dimensions relate to subjective measures (WNSS, SSQ, ANL) versus an objective test (DTT) of hearing in noise (figure 2 and table 1).

Figure 3 visualizes this pattern of results and shows normalized single-subject scores of subjective hearing in noise as a function of objective hearing in noise, weighted by neuroticism (a), by extraversion (b) and—for comparison—by age (c). Individuals expressing better subjective than objective hearing in

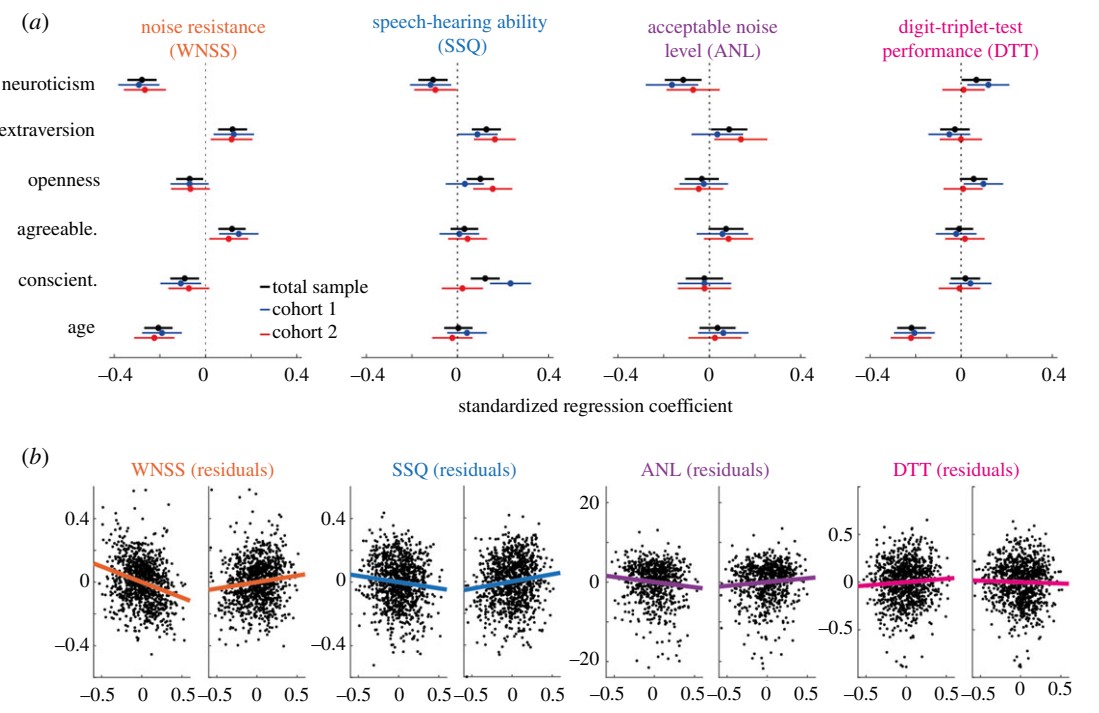

**Figure 2.** Associations of Big-5 personality dimensions and age with hearing in noise. (*a*) Dots and horizontal lines show respective standardized coefficients and 95% confidence intervals, resulting from four multiple linear regression models to regress scores of hearing in noise tests on personality dimensions and demographic information. Positive versus negative coefficients indicate that higher scores of the predictor under consideration relate to higher versus lower hearing in noise scores, respectively. Black, blue and red lines show results for the total sample, cohort 1 and cohort 2, respectively. The number of observations for each multiple regression were: WNSS (total: 1015; cohort 1: 511; cohort 2: 504), SSQ (total: 1031; cohort 1: 522; cohort 2: 509), ANL (total: 723; cohort 1: 367; cohort 2: 356) and DTT (total: 1000; cohort 1: 506; cohort 2: 494). Note that categorical predictors (gender, education, sound presentation and hearing loss) are omitted here but shown in table 1. (*b*) Scatterplots show hearing in noise scores (residuals from the multiple regression models) as a function of neuroticism and extraversion (residuals). Residuals were derived by regressing out all other predictors from each one of the two variables included in the respective scatterplot.

noise can be considered to overrate their hearing in noise, while the opposite (underrated hearing in noise) holds for individuals expressing better objective than subjective hearing in noise.

Notably, higher neuroticism predicted lower scores of subjective, but higher scores of objective hearing in noise. The reverse pattern was observed for extraversion. Compared with surrogate data with permuted personality scores, neuroticism and extraversion significantly displaced participants from an equilibrium of subjective and objective hearing in noise. That is, higher scores on neuroticism were accompanied by underrated own hearing in noise. Instead, higher scores on extraversion were associated with overrated hearing in noise. By contrast to personality, higher age reduced objective hearing in noise but left the subjective hearing in noise largely unaffected.

For statistical analysis, we employed a new joint extraversion–neuroticism predictor in additional multiple regression analyses: the normalized difference $extraversion_{z\text{-score}} - neuroticism_{z\text{-score}}$ explained inter-individual differences in subjective hearing in noise (WNSS: $\beta = 0.326$; $p < 0.001$, CL = 0.604; SSQ: $\beta = 0.193$; $p < 0.001$; CL = 0.562; ANL: $\beta = 0.166$; $p < 0.001$; CL = 0.552). This effect corresponds to a considerable displacement along the vertical axis in figure 3*d*. Although smaller in effect size, the extraversion–neuroticism difference score also explained objective hearing in noise (DTT: $\beta = -0.077$; $p = 0.0124$; CL = 0.475), corresponding to a small displacement of participants along the horizontal axis in figure 3*d*. In other words, relatively higher extraversion and lower neuroticism explain the tendency to subjectively overrate one's objective hearing in noise.

To further illustrate this dissection along the personality dimensions neuroticism (N) and extraversion (E), figure 3*e* shows median splits along both personality dimensions that divide participants with scores below the median (−) from participants with scores above the median (+).

Beyond personality traits neuroticism and extraversion, table 1 shows that other BIG-5 personality traits explained variance of hearing in noise tests. Higher self-reported noise resistance in the WNSS

**Table 1.** Test statistics (t), statistical significance (p) and effect sizes (r) from multiple regression analyses. Rows correspond to predictor variables; columns correspond to the four outcome measures of hearing in noise. WNSS = Weinstein noise sensitivity scale; SSQ = speech spatial and qualities scale; ANL = acceptable noise level; DTT = digit-triplet-test; speak = speakers; in/ex = internal (in-built)/external (tabletop); headp = headphones; highest school-leaving qualification (from lowest to highest): main, mid sch = school, AVCE = advanced vocational certificate of education (German 'Fachabitur'); A-lev = A-levels.

| | WNSS | | | SSQ | | | ANL | | | DTT | | |
|---|---|---|---|---|---|---|---|---|---|---|---|---|
| | t | p | r | t | p | r | t | p | r | t | p | r |
| neuroticism | −8.590 | <0.001 | −0.262 | −3.287 | 0.001 | −0.103 | −2.792 | 0.005 | −0.104 | 2.005 | 0.045 | 0.064 |
| extraversion | 3.683 | <0.001 | 0.116 | 3.925 | <0.001 | 0.122 | 2.127 | 0.034 | 0.080 | −0.843 | 0.399 | −0.027 |
| openness | −2.308 | 0.021 | −0.073 | 3.313 | 0.001 | 0.103 | −0.870 | 0.384 | −0.033 | 1.770 | 0.077 | 0.056 |
| agreeableness | 3.813 | <0.001 | 0.120 | 0.995 | 0.320 | 0.031 | 1.874 | 0.061 | 0.070 | −0.288 | 0.773 | −0.009 |
| conscientious | −2.852 | 0.004 | −0.090 | 3.767 | <0.001 | 0.117 | −0.527 | 0.598 | −0.020 | 0.551 | 0.582 | 0.018 |
| age | −6.572 | <0.001 | −0.203 | 0.130 | 0.897 | 0.004 | 0.891 | 0.373 | 0.033 | −6.815 | <0.001 | −0.212 |
| male versus female | 1.133 | 0.258 | 0.036 | −2.966 | 0.003 | −0.093 | 1.673 | 0.095 | 0.063 | 1.990 | 0.047 | 0.063 |
| in speak versus headp | 1.006 | 0.315 | 0.032 | −0.534 | 0.593 | −0.017 | −1.167 | 0.244 | −0.044 | −8.003 | <0.001 | −0.247 |
| ex speak versus headp | 1.955 | 0.051 | 0.062 | −0.305 | 0.761 | −0.010 | −1.957 | 0.051 | −0.073 | −5.043 | <0.001 | −0.159 |
| AVCE versus A-lev | 3.289 | 0.001 | 0.103 | 1.547 | 0.122 | 0.048 | −1.404 | 0.161 | −0.053 | −1.213 | 0.225 | −0.039 |
| mid sch versus A-lev | 1.526 | 0.127 | 0.048 | −2.700 | 0.007 | −0.084 | −0.529 | 0.597 | −0.020 | 0.355 | 0.723 | 0.011 |
| main sch versus A-lev | 1.567 | 0.117 | 0.049 | −0.205 | 0.838 | −0.006 | −3.144 | 0.002 | −0.117 | −0.475 | 0.635 | −0.015 |
| hearing loss | 0.273 | 0.785 | 0.009 | −5.795 | <0.001 | −0.179 | −0.614 | 0.539 | −0.023 | −1.961 | 0.050 | −0.062 |

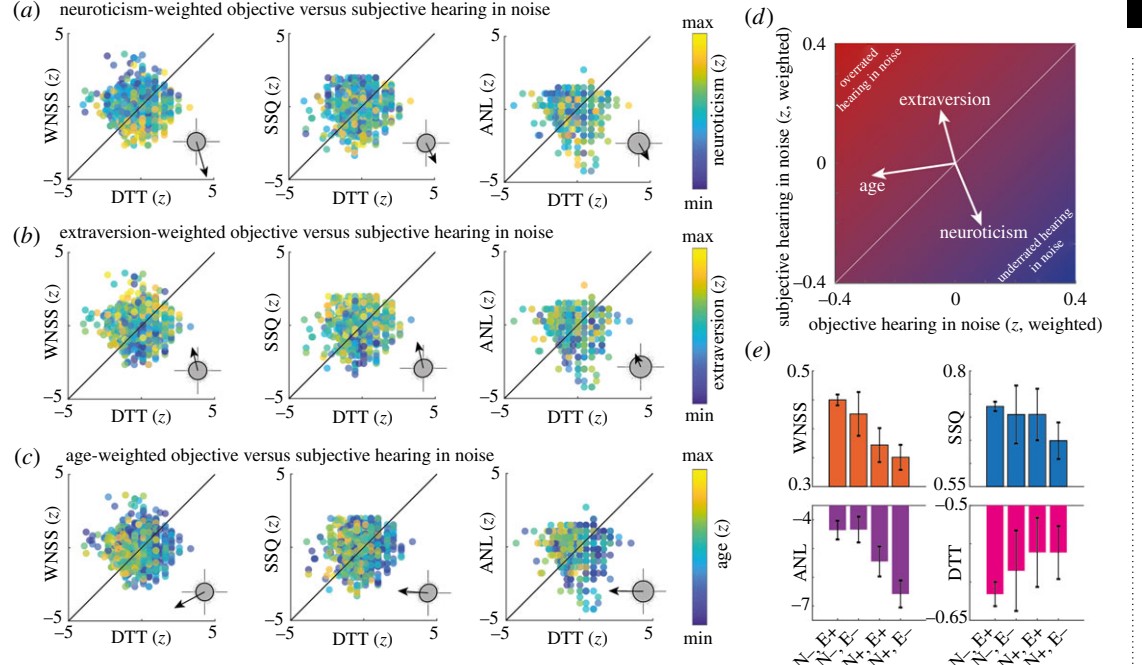

**Figure 3.** Personality explains objective versus subjective hearing in noise. (a) Scatterplots show normalized (z-scored) subjective hearing in noise (for WNSS [left], SSQ [middle], ANL [right]; y-axes) as a function of objective hearing in noise (DTT; x-axes). Dots show individual participants. Warmer colours indicate higher scores on neuroticism. Arrow heads in the insets show centres of gravity of neuroticism-weighted dot clouds. Grey dots and black circles show centres of gravity for 10 000 permutations of neuroticism scores across participants and the corresponding 95% elliptical confidence intervals, respectively. Axes in the insets are scaled between −0.2 and +0.2. (b,c) The same as a but for extraversion and age. (d) Summary of results shown in (a–c). Arrow heads point to centres of gravity of dot clouds averaged across the three subjective measures of hearing in noise (WNSS, SSQ, ANL). (e) Participants were divided into four groups by median split (+, above median; −, below median) of neuroticism (N) and extraversion (E). Numbers of participants for each group were as follows: N−,E+ (250), N−,E− (128), N+,E+ (166), N+,E− (274). Bars and error bars show mean and SEM of scores on hearing in noise tests, respectively.

was associated with higher agreeableness ($\beta = 0.116$; $p < 0.001$; CL = 0.538) but with lower openness ($\beta = -0.07$; $p = 0.021$; CL = 0.477) and conscientiousness scores ($\beta = -0.091$; $p = 0.004$; CL = 0.471). Furthermore, better self-reported speech-hearing ability in the SSQ was associated with higher openness ($\beta = 0.101$; $p < 0.001$; CL = 0.533) and conscientiousness scores ($\beta = 0.122$; $p < 0.001$; CL = 0.537).

# 3. Discussion

Humans differ considerably in their tolerance of acoustic noise in the environment. Here, we investigated the association of Big-5 personality traits neuroticism and extraversion with subjective and objective measures of hearing in noise. We used an online experiment to recruit a large, age-varying sample of participants ($N = 1103$; 18–74 years) and employed statistical control of demographics and other personality traits. Higher neuroticism and lower extraversion were associated with substantially reduced subjective hearing in noise but with somewhat increased objective hearing in noise. These relations were largely orthogonal to the effect of older age, which primarily explained decreasing objective hearing in noise.

## 3.1. Personality explains multiple facets of subjective hearing in noise

Our results replicate well-established associations of neuroticism and extraversion with subjective (i.e. self-reported) noise tolerance on the WNSS [5]. Our effect size estimates indicate the following: if, for example, we picked two women of same education, same technical set-up for the listening task and same self-reported hearing loss from our sample, and one of them were known to be the less emotionally stable (i.e. N+) of the two—then we would expect her to also be self-reportedly (on the WNSS) the less noise tolerant with an approximately 58% probability. Likewise, if we randomly had picked two participants who only differed in their extraversion score, our results would make us

expect the more extraverted of the two to be the self-reportedly more noise tolerant (approx. 54% probability).

The relationship of personality and scores on the WNSS are unsurprising, given that this scale was designed to assess affective reactions to noise [5] and that individual items probe the degree of (noise-induced) disturbance, irritation and stress, all of which characterize unstable introverts (high neuroticism, low extraversion).

A major goal of the present study was to test whether neuroticism and extraversion would also explain scores on hearing in noise tests used in audiological screening and hearing aid fitting. Our results clearly showed that this was the case: Higher neuroticism and lower extraversion independently explained lower self-reported speech-hearing ability (SSQ; [31]) and lower self-adjusted levels of noise (see also [32]) when listening to speech (ANL; [33]).

The difficulty of speech comprehension in noise is a common complaint, which has primarily been associated with older age, hearing loss [34] and lower cognitive capacity [35]. Here, we demonstrate that personality screening helps understanding individual differences of reported hearing in noise and speech-hearing ability, which is a long-standing and unresolved challenge in audiology [36–38]. To appreciate to what extent personality is associated with actual hearing in noise or with the self-evaluation thereof, it is necessary to relate personality to objective measures of hearing in noise.

## 3.2. Personality can account for objective hearing in noise

The DTT is a widely used audiological screening tool available in many languages [39]. It is administered in the laboratory, but also over the telephone [40] or Internet [41]. In the case that the background noise is non-speech (as in the present study), it does not constitute a meaningful signal that would draw attention away from the primary task of digit recognition. Research suggests that the DTT primarily assesses hearing sensitivity [42,43] and that associations with non-auditory factors such as working memory or attention are considerably smaller [44]. In this sense, the DTT constitutes a well-established, objective test of hearing in noise.

In the present study, better objective hearing in noise was to large extents explainable by younger age and use of headphones (versus tabletop or in-built loudspeakers). Opposite to our hypotheses, neuroticism exhibited a small positive association with objective hearing in noise under statistical control for demographic information and sound presentation as well as other personality traits: if a person sampled at random scored above average on neuroticism, the probability of their objective hearing in noise being also above average is about 52%. It is obvious that the association of DTT scores with neuroticism was considerably smaller than their association with age, which is known to covary with hearing thresholds. Since hearing sensitivity (i.e. audiometry) was not included in the present study, we cannot differentiate whether higher neuroticism relates to lower (i.e. better) hearing thresholds or to better hearing in noise, which are both captured by the DTT.

Together, the differential relations of neuroticism with subjective versus objective hearing in noise have important implications for audiological rehabilitation: it might be instructive to imagine an audiologist who encounters over a few months' time or so 100 customers with above-average neuroticism. Based on the present results, the audiologist would expect 63 of them to also be of sub-average self-reported hearing in noise (and not 50)—but should not be surprised if 53 of them in fact perform above average in an objective hearing in noise test [45]. Note that what appears to be a relatively small effect can amount nevertheless to considerable real-life relevance: humans are confronted with plenty of speech-in-noise situations in everyday life, such that even relatively minute associations with personality cumulate over time [46,47].

Diverse associations between neuroticism and extraversion on the one hand and sensory processing on the other have been reported in the literature. For instance, lower extraversion correlates with lower (i.e. better) auditory thresholds [32,48] and unstable introverts were found to exhibit lower electrocutaneous stimulation thresholds [49]. Contrary, higher neuroticism relates to increased distraction by task-irrelevant stimuli [50,51]. What are the underlying mechanisms that explain how neuroticism relates to reduced subjective but enhanced objective hearing in noise?

## 3.3. Relation of neuroticism with subjective versus objective hearing in noise

High levels of neuroticism have traditionally been considered a predisposition for negative affect [52–54], which explains reduced (i.e. more pessimistic) evaluation of subjective hearing in noise. Furthermore, implications of neuroticism for cognition have been suggested [55]. There is evidence that the automatic

orienting of attention is increased for individuals with higher levels of neuroticism, which might explain increased levels of distraction by task-irrelevant stimuli [56]. But which mechanism can account for better objective performance in a hearing in noise test in listeners with higher levels of neuroticism?

In theory, individuals with poorer subjective hearing in noise might choose better hardware for sound presentation, which in turn might improve objective measures of hearing in noise. This should result in a negative correlation of subjective and objective hearing in noise, which was not the case in the present data. Furthermore, our statistical models did control for quality of sound presentation to some extent (headphones versus in-built or tabletop speakers). However, a caveat of the present online study was the unfeasibility to control the level of ambient noise settings in which individual participants performed the study.

We consider three views that explain the neuroticism-objective hearing in noise relationship. First, the view that unstable introverts (high neuroticism, low extraversion) are characterized by higher levels of neural excitation and lower inhibition [57–59] and second, the view that mental noise increases with neuroticism [60], indicated by e.g. increased response time variability [61]. However, higher excitation in response to the acoustic signal, which is dominated by noise in the DTT, and mental noise should reduce rather than enhance DTT performance. Thus, none of these views can explain the positive relation between neuroticism and objective hearing in noise.

Third, and more promisingly, individuals with higher neuroticism might allocate more cognitive resources [62], which eventually results in better performance in tasks that require high effort [63]. Some positive associations were found for neuroticism and self-reported [64], as well as physiological measures of effort [65]. Of note, evidence for the direction of the association of neuroticism and effort is mixed. For instance, the existing literature also includes studies that found no association of neuroticism and auditory vigilance [66], as well as evidence that separate aspects of neuroticism might relate differently to task performance [67].

In the context of the hearing, the allocation of cognitive resources is referred to as *expended listening effort* or *listening engagement* [68–70]. In the second cohort of the present study, we included two questions following the DTT to probe participants' experienced mental demand and expended effort. Under statistical control for potential confounders (age, gender, sound presentation, highest education, hearing loss), neuroticism was a significant positive predictor of expended effort ($\beta = 0.099$; $p = 0.031$) but not of experienced mental demand ($\beta = 0.077$; $p = 0.091$). These results are consistent with the view of higher expended listening effort being associated with higher levels of neuroticism.

Reminiscent of trait neuroticism, the listening effort construct has been related to both affect and cognition [71]. Recently, initial attempts were made to explore the relation of personality to listening effort [72,73]. Mechanistically, the present results might suggest that individuals with higher levels of neuroticism do not have superior hearing in noise abilities *per se*, but rather that they allocate more cognitive resources, which eventually results in lower speech reception thresholds in the DTT.

Of note, objective hearing in noise was significantly associated with only one of the three subjective hearing in noise tests (SSQ). This agrees with and extends the literature on discrepancies between subjective and objective hearing ability [17,19,20]. However, discrepancies between hearing in noise tests might be somewhat overestimated here since the noise was operationalized differently (WNSS: items asking for various types of noise in everyday life; SSQ: items asking for speech-hearing ability in multi-talker noise; ANL: tolerable level of multi-talker noise while listening to target speech; DTT: target speech reception in broadband noise).

As a direct consequence of the differential relation of personality with subjective versus objective hearing in noise, higher neuroticism associates with a tendency to underrate one's own hearing in noise, whereas the opposite holds for higher extraversion. Put differently, subjective and objective hearing in noise tests are of limited validity: the former might be sensitive to affect and the latter to the allocation of cognitive resources (i.e. listening effort), both of which vary as a function of neuroticism. Future research should (i) explore to what degree the link between neuroticism and performance in objective hearing in noise tasks is mediated by listening effort, (ii) test whether the observed associations with personality generalize to non-auditory sensory processing sensitivity [74] and (iii) elucidate which audiological screening tests minimize associations with personality.

It is important to emphasize that the relation of personality with subjective hearing in noise was considerably larger than the association with objective hearing in noise. In this sense, a straight-forward interpretation of the present study is that personality traits neuroticism and extraversion predict how well listeners feel they do in noise or how well they can deal with noise. By contrast, listeners' objective hearing sensitivity or hearing in noise show a smaller, less well-defined association with personality.

## 3.4. Is personality causal for hearing in noise, and does it matter?

Our goal here has been to estimate, in a large sample of participants, the direction and magnitude of the long-hypothesized association of neuroticism and extraversion on the one hand and subjective and objective measures of hearing in noise on the other. We did so under statistical control of reasonable potential confounds like gender, age, education and hearing loss. Therefore, we think we have captured the true association with satisfactory precision.

One might argue that there are not many obvious, unmeasured confounds left that would be able, if measured, to explain away this association in its entirety. Nevertheless, there is no strong evidence presented here to argue for either direction of causality between personality and hearing in noise. While a causal effect of relatively stable personality traits on situational aspects of hearing in noise is more parsimonious, the reverse causality should not be disregarded: for example, low levels of tolerance to noise or overall sensory hypersensibilities early in life might well have an impact on personality development [75,76].

Irrespective of these unsolved causal questions, we think that the demonstrated association is one of the rare cases where even a correlative result, devoid of a clear mechanistic pathway, will be informative and of potential relevance to future environmental psychology, public health and audiological treatment decisions: Even if personality is a causal factor for hearing in noise, it is not an easily manipulable target for changes [77–79].

In practice, short personality inventories might be used to obtain knowledge about the likelihood of a specific individual to under- or over-estimate their own hearing in noise, relative to other individuals. This could pose valuable clinical information and help manage client expectations in the extended period of fitting a new hearing device to an individual. For instance, if an individual scores particularly high on neuroticism, it is worth considering the possibility that this individual might tend to underrate hearing in noise abilities. This might also help to explain potential dissatisfaction with a hearing device.

However, it must be noted that although small associations of personality and hearing in noise probably cumulate over time in real life, it should not be expected that audiologists detect an obvious association of personality and a single audiological screening test on the level of the individual client. At the present stage, the most important conclusion of this study for clinical audiology is that audiologists and their clients should be sensitized to the fact that subjective and objective measures of hearing in noise often dissociate, and that personality explains part of this dissociation.

# 4. Conclusion

The present study reveals differential associations of the key personality traits neuroticism and extraversion with established subjective versus objective test of hearing in noise. Results first emphasize the importance of considering listeners' personality in the interpretation of scores resulting from audiological screening tests. Second, they set the stage for future studies to test the hypothesis that neuroticism relates to reduced subjective hearing in noise via negative affect and to enhanced objective hearing in noise via the increased expenditure of listening effort. Third and of direct relevance to public health, these data pose robust evidence from a large sample that explain individual differences in self-reported and objectively assessed hearing in noise from non-audiological predictors of individual behaviour—namely, personality.

# 5. Methods

## 5.1. Online experiment

The present experiment was implemented using the online platform *Labvanced* (http://www.labvanced.com). Participants completed all tests of the experiment in the browser. After piloting the functionality of the online study, all procedures were pre-registered with the Open Science Framework before data collection started (osf.io/fgyj9).

To make sure that a participant's computer was playing audio files at an appropriate sound intensity, each participant was presented with a spoken German digit triplet and had to select the corresponding triplet among six options in the beginning of the online study. The study continued only in case this task was successfully accomplished.

## 5.2. Participants

Participants were recruited via crowdsourcing. The study was only available to participants who, according to their registration with the crowdsourcing platform, were German. They provided informed consent and were financially compensated by €3.60 if they completed all tests of the study. Experimental procedures were approved by the ethics committee of the University of Lübeck.

Data were collected in two cohorts. The planned sample size in cohort one was at least $N = 500$. To be able to replicate the obtained results, the planned sample size in cohort two was also at least $N = 500$. Participants who completed at least the demographic questionnaire and one additional test were included in the final sample, which were $N_{cohort\ 1} = 554$ recorded in December 2019 and January 2020, and $N_{cohort\ 2} = 549$ recorded in April and May 2020, resulting in a complete sample of $N_{total} = 1103$ (463 females; 637 males; 3 diverse; age range: 18–74 years; mean age: 38.6 years; 32 with self-reported hearing loss).

Every participant completed the questionnaire on demographic information first, followed by the remaining five tasks (personality questionnaire, BFI-S; noise resistance questionnaire, WNSS; speech-hearing ability questionnaire, SSQ; acceptable noise level test, ANL; digit-triplet-test, DTT) in randomized order to avoid order effects.

— *Demographic information*. Participants provided information on their gender (male/female/diverse), age, number of siblings, education (highest school-leaving qualification and number of years of school education), musicality (number of years playing a musical instrument and start age of playing a musical instrument), knowledge of existing hearing loss (yes/no), use of hearing prostheses (no/hearing aid/cochlear implant/other) and the type of sound presentation of their computer (internal speakers/external speakers/headphones). Of note, no additional direct measures of hearing sensitivity (e.g. mobile-based audiometry; [80]) were included in the present online study in order to keep the overall study duration short. Furthermore, online audiometric tests would have required additional validation procedures, such as comparison of online with laboratory audiometry for a representative sub-sample of subjects.

— *Personality questionnaire*. Big-5 personality dimensions (neuroticism, extraversion, openness, agreeableness and conscientiousness) were assessed using the short BFI-S questionnaire [30,81], composed of three items per dimension. To normalize the possible range of scores between 0 and 1, item scores per dimension were summed up and divided by the highest possible score.

— *Noise resistance*. Noise resistance was assessed using a German version [82] of the WNSS [5], composed of 21 items on a 6-point scale. To normalize the possible range of scores between 0 and 1, item scores were summed up and divided by the highest possible score. In order that higher scores reflect higher noise resistance (i.e. the opposite of noise sensitivity), scores were subtracted from 1.

— *Speech-hearing ability*. Six items of the speech, spatial and qualities questionnaire (SSQ; items Speech 1, 2, 3, 6, 7 and 10 from [31]; translated to German according to Kießling *et al.* [83]) were used to assess self-rated speech-hearing ability under challenging conditions. The range of possible scores (0–10) was scaled between 0 and 1, with higher scores reflecting better speech-hearing ability.

— *ANL test*. We implemented an adapted version of the ANL test [33]. Participants listened to 10 s snippets of a German audiobook (*Eine kurze Geschichte der Menschheit*; Yuval Noah Harari) narrated by a male voice, which was embedded in 10-talker babble noise (adopted from [84]). Using two buttons in the browser, participants could adjust the signal-to-noise ratio (SNR) in steps of 2 decibels (dB). To make sure that participants experienced a considerable range of SNRs, participants first had the task to adjust the SNR so that the background noise was too loud (loud noise level; LNL). Next, they adjusted the SNR in order that the audiobook was clearly intelligible (soft noise level; SNL). Finally, participants selected an SNR that was just acceptable (ANL). Obtained mean SNR levels in this test were $M_{SNL} = 11.86$ dB, $M_{LNL} = -1.80$ dB and $M_{ANL} = 5.20$ dB. In order that higher ANL values reflect higher acceptable noise levels, ANL scores were sign-flipped (i.e. multiplied by $-1$).

— *DTT*. To obtain an objective measure of speech-in-noise reception, we used an adapted version of the DTT [85]. On every trial, participants were presented with a German digit triplet, composed of three random numbers from the interval [0;9]. Triplets were spoken by a female voice (e.g. '5 … 3 … 9') and embedded in stationary speech-shaped noise (each triplet was embedded in a different random section from a longer noise signal), which started 500 ms before the onset of the first spoken digit and ended 500 ms after the offset of the last spoken digit. Participants had the task to select the

three digits of the triplet on a visually presented number pad. Using an adaptive tracking one-up, one-down procedure, we estimated a participant's SNR corresponding to 50% correct task performance, referred to as speech reception threshold ($SRT_{50}$). To this end, the test started at an SNR of 0 dB. For every subsequent trial, the SNR was raised by 2 dB in case of an incorrect response on the previous trial and decreased by 2 dB in case of a correct response on the previous trial. Participants completed 24 trials. The $SRT_{50}$ was calculated as the average SNR across the last 6 trials. Participants completed two runs of the DTT, with mean SNR values of $M_{DTT1} = -11.03$ dB and $M_{DTT2} = -11.42$ dB.

To better agree with the assumption of normality, obtained scores ($DTT_{raw}$) were shifted into the range of positive values and log-transformed using the formula: $DTT = \log_{10}(DTT_{raw} + 16)$ for further statistical analyses. Since previous research has shown that performance in the DTT reaches stable performance levels in the second run of the test [85], we used scores of the second run as the DTT score for all further analyses. To reflect that higher DTT values denote better speech-in-noise reception, DTT scores were sign-flipped (i.e. multiplied by −1).

After the DTT, participants in cohort two completed two short questionnaires to assess some aspects related to listening effort. Two questions (adapted from the NASA-task load index; [86]) were intended to assess mental demand ('How high was your mental demand for this listening test?'; translated from German) and expended effort ('How hard did you have to work to reach your level of performance?'; translated from German) on a 10-point scale (1, very low; 10, very high).

## 5.3. Data pre-processing

A table including pre-processed data and an accessible script for most important data analyses in the open access software Jamovi (http://www.jamovi.org) are available online (osf.io/fgyj9). Further analysis scripts and raw data are available from the corresponding author upon request.

Data pre-processing was implemented in Matlab [87]. Critical for an online study with limited control of how participants accomplish the intended tasks, we carefully validated the quality of the obtained data. For every questionnaire in the present study, we included up to two control items (e.g. 'Please select the leftmost response') to make sure that participants read and responded to the items in a sensible way. If a participant's responses to one or more of these control items was incorrect, this participant's scores on the respective questionnaire were removed.

Since the sample included only three participants with diverse gender, gender information for these participants was classified as missing data. For the DTT, data of participants with speech reception thresholds greater than +15 dB were considered implausibly high and removed. The strictest exclusion criteria were applied to the ANL test, where an ANL was only considered valid if it was in-between the SNL and the LNL and removed otherwise. After applying these exclusion criteria to the data of a total sample of $N_{total} = 1103$, the following number of participants remained for individual tests: $N_{personality} = 1049$; $N_{WNSS} = 1043$; $N_{SSQ} = 1075$; $N_{ANL} = 742$; $N_{DTT} = 1031$.

To validate individual tests of the online study, internal consistency was assessed with CA for scales and with test–retest reliability for the DTT (i.e. Pearson correlation of first and second run of DTT).

## 5.4. Statistical analyses and effect sizes

We used two-sided statistical test. Planned statistical analyses were supposed to test the relation of personality dimensions and different facets of hearing in noise. To this end, bivariate Pearson correlations and multiple linear regression analyses were performed as implemented in Jamovi (v. 1.1.9) and in Matlab [87]; using the *fitlm* function). Standardized coefficients were obtained by z-transforming all variables in the regression model. The 95% CIs for all effects (i.e. standardized regression coefficients) can be found in the online data (osf.io/fgyj9).

The major statistical analysis comprised four multiple regression models (figure 2). To control for false positives, we did not correct $p$-values for the number of performed tests but instead tested the replicability of all effects by splitting up the dataset into the two cohorts. Bivariate Pearson correlations among predictors in regression models were small to moderate (all $|r| < 0.35$; see also figure 1$b$) and the variance inflation factors of regression models were small (all VIF < 2), such that multicollinearity was not an issue.

Missing values on individual tests were not imputed. Thus, any case (participant) with a missing value was discarded from the respective correlation analysis or regression model. The number of

observations for the four multiple regression models (with outcome measures WNSS, SSQ, ANL and DTT) were: WNSS (total: 1015; cohort 1: 511; cohort 2: 504), SSQ (total: 1031; cohort 1: 522; cohort 2: 509), ANL (total: 723; cohort 1: 367; cohort 2: 356) and DTT (total: 1000; cohort 1: 506; cohort 2: 494).

To estimate the internal consistency of the relation of neuroticism and performance in the DTT (in the multiple linear regression analysis), we used a sub-sampling approach. We drew 10 000 sub-samples (each with $N = 550$, sampled at random without replacement) from the whole sample of participants, followed by the calculation of a multiple linear regression to predict DTT performance. Standardized coefficients for the relation of neuroticism and DTT performance were extracted ($\beta_{\text{sub-sample}}$) and corresponding bounds of the 95% CI were estimated by the 2.5th and 97.5th percentiles of the distribution of $\beta_{\text{sub-sample}}$.

To derive effect sizes for individual predictors in multiple regression models, we calculated the standardized partial effect size $r$ according to the equation [88]

$$r = \sqrt{\frac{t^2}{t^2 + df}}. \tag{5.1}$$

The resulting $r$ value was multiplied by the sign of the respective $t$-value to reflect the direction of the effect.

To further ease the interpretation of effect sizes, we report the common language (CL) effect size [89], which was generalized to bivariate normal correlations by Dunlap [90], according to the equation

$$\text{CL} = \frac{\sin^{-1}(r)}{\pi} + 0.5. \tag{5.2}$$

For instance, for a correlation of the variables $A$ and $B$, a CL of 0.66 means the following: for a randomly sampled person, given that this person has an above-average score on variable $A$, the probability of an above-average score on variable $B$ is 66%. For tests that were most central to our major hypotheses, we calculated the CL for $r$-values resulting from bivariate correlations and for the standardized partial effect size $r$ from multiple regression analyses.

Furthermore, we performed unplanned analyses to statistically test the relation of personality with diverging scores on subjective versus objective hearing in noise. To this end, we calculated normalized scores by z-transform of hearing in noise test scores (WNSS, SSQ, ANL, DTT), personality dimensions neuroticism and extraversion, and age. For purpose of visualization (figure 3a–c), we generated scatterplots of subjective hearing in noise (WNSS, SSQ, ANL) as a function of objective hearing in noise (DTT), followed by colour-coding of individual participants' data points by neuroticism, extraversion and age. For statistical analysis, we permuted neuroticism, extraversion and age across participants (10 000 permutations), followed by calculation of the weighted centre of gravity (COG) of data in the objective-versus-subjective hearing in noise space. For instance, the $x$- and $y$-coordinates of the neuroticism-weighted COG of objective (DTT) versus subjective (WNSS) hearing in noise was calculated according to

$$\text{COG}x = \frac{1}{N}\sum_{s=1}^{N} \text{DTT(s)} * \text{neuroticism(s)} \tag{5.3}$$

and

$$\text{COG}y = \frac{1}{N}\sum_{s=1}^{N} \text{WNSS(s)} * \text{neuroticism(s)} \tag{5.4}$$

Elliptical 95% CIs of surrogate COGs were calculated using the function *error_ellipse* for Matlab [91].

Ethics. Participants were recruited via crowdsourcing. They provided informed consent and were financially compensated by €3.60 if they completed all tests of the study. Experimental procedures were approved by the ethics committee of the University of Lübeck.

Data accessibility. A table including pre-processed data and an accessible script for data analyses in the open access software Jamovi (http://www.jamovi.org) are available online ([92]; https://osf.io/fgyj9/; doi:10.17605/OSF.IO/FGYJ9).

Authors' contributions. M.W. and J.O. conceptualized the study design. M.W. collected and analysed the data. J.O., J.E. and J.K. contributed substantially to data analyses and statistical analyses. All authors contributed to writing the manuscript.

Competing interests. The authors declare no conflict of interest.

Funding. The present work was supported by the International Hearing Foundation (grant to M.W. and J.O.) and an ERC Consolidator grant (ERC-CoG-646696 AUDADPT) to J.O. We acknowledge financial support by the Land Schleswig-Holstein within the funding programme Open Access Publikationsfonds.

Acknowledgements. We thank Felix Greuling for implementing the online study.

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
