## [Peer Review File · Royal Society Open Science]

Review History

RSOS-210881.R0 (Original submission)

Review form: Reviewer 1 (Alexander L. Francis)

Is the manuscript scientifically sound in its present form?

Yes

Are the interpretations and conclusions justified by the results?

Yes

Is the language acceptable?

Yes

Do you have any ethical concerns with this paper?

No

Have you any concerns about statistical analyses in this paper?

No

Recommendation?

Accept with minor revision (please list in comments)

Comments to the Author(s)

I think this is an interesting and useful study that seems to have been conducted quite well to the extent that I am able to evaluate it without being familiar with the statistical methods employed here. Despite my lack of experience with these methods, I think they are sufficiently well described for an interested and determined reader to follow up on adequately. Most importantly, the methods and results are presented with sufficient clarity to permit replication (and the availability of the preregistration, data, and analysis scripts is very helpful in this regard). I think the overall results contribute to our understanding of individual differences in noise sensitivity, and the exploratory analyses provide a basis for some intriguing avenues for future research. I have only a few suggestions for small changes that I think will improve the readability of the manuscript. (note - page numbers are based on those assigned by the editorial manager software, not the ones provided by the authors in the ms)

p. 17

around line 20 – Based on data from OSF it looks like school qualifications were from the German school system. Were participants all from Germany? From German-speaking regions? How was this ensured? A great deal of emphasis is put on the rigor of demographic selection, but not much is described in the methods.

Around line 33 – it seems as if the final analyses could have been done on either the complete sample N_{total} with missing values compensated for somehow, or on the sample that completed all tasks ($N_{total}-61$) but it's not clear which was the decision. As this is explained in a little more detail on p. 19, maybe it makes sense to just report the overall total here, and then go into the full detail in the data pre-processing section (especially since I don't see the $N=1042$ group in the data pre-processing explanation anyway).

around line 35 – I would recommend listing the 5 tasks here. It's a little redundant, but it will help readers who have skipped straight to the methods (as one does).

p. 18

around line 3 – I think it would be clearer to say that the ratio scores were then subtracted from 1, rather than "inverted" If I am correct that the score is $1 - (\text{score}/(21*6))$?

Around line 12 – please give detailed methods as in the Noise resistance score (to make it easy to replicate accurately). Which six items? What were the max scores for each item, etc.

Around line 21 – was there any way to ensure that participants were in fact adjusting the levels as requested? If not, how was the reliability of these measures ensured? Was there anything like the "catch trials" in the surveys?

Around line 38 – why were ANL scores not normalized to between 0 and 1 as the others were? Would having these scores be on a much larger and less-fine-grained scale affect the statistical analyses (i.e. the other variables in the mix range from 0 to 1 in 0.01 unit steps, while ANL scores seem to vary from -26 to 8 in steps of 2). (DTT ranges from -1.49 to 0.17 but there are only 31 unique values).

P 20

Around line 7 – Given that all subjects with missing data were excluded from that analysis (but, I think, not from other analyses for which they had given responses) I think it makes sense to

report numbers of subjects as I described in the comment re. p. 17 – and I see that this was done nicely in Figure 2.

P 10

Around lines 50-55 I find the sentence “That is, higher scores on extraversion versus neuroticism were accompanied by overrated versus underrated own noise tolerance, respectively” to be very confusing. I would recommend unpacking it into multiple sentences, as this seems to be a crucial finding and one that might easily be missed or misunderstood from this phrasing.

P 12

Line 33 – “extravert” should be “extraverted”

Review form: Reviewer 2

Is the manuscript scientifically sound in its present form?

Yes

Are the interpretations and conclusions justified by the results?

No

Is the language acceptable?

Yes

Do you have any ethical concerns with this paper?

No

Have you any concerns about statistical analyses in this paper?

Yes

Recommendation?

Major revision is needed (please make suggestions in comments)

Comments to the Author(s)

RSOS-210881

The authors conducted a large scale study of the relationship(s) between personality traits and 4 measures of hearing/listening [Weinstein Noise Sensitivity Scale (WNSS), the Acceptable Noise Level (ANL) test, the Speech items of the Speech, Spatial and Qualities (SSQ) scale and the Digits-Triplet-Test (DTT)]. The results suggest that different personality traits (neuroticism, and to a lesser extent extraversion) can impact results from hearing/listening tests.

I have some concerns over the interpretation of results (please see below).

1) Interpretation

1a) Outcomes described as measures of “noise tolerance”

I understand why the authors would want to simplify their findings and categorize all 4 outcome measures as measures of “noise tolerance”. I agree with the authors that 2 of the tests (Weinstein Noise Sensitivity Scale (WNSS) and the Acceptable Noise Level (ANL)) are tests of noise tolerance. But I am not clear on why the authors also describe the Speech items of the Speech, Spatial and Qualities (SSQ) scale and the Digits-Triplet-Test (DTT) as “noise tolerance tests”.

The authors describe the selected items of the SSQ (“Speech-hearing items”; Gatehouse and Noble, 2004) as a measure of “speech-in-noise comprehension”. Gatehouse & Noble used the term “speech-hearing ability” to describe what the SSQ measures.

The DTT is not a noise tolerance test. It can be used as a test of hearing acuity or screen for hearing loss e.g. the World Health Organisation’s hearWHO app. Indeed, on page 12 the authors state that the DTT primarily assesses hearing sensitivity.

The authors should revisit their descriptions of the SSQ and the DTT as “noise tolerance tests” (in the title and throughout).

1b) Interpretation re listening effort

On page 12 the authors argue that non-auditory factors such as memory or attention have a negligible impact on DTT performance. Then on page 13 the authors argue that people who are higher in neuroticism allocate more cognitive resource which “eventually results in lower speech reception thresholds in the DTT”. These statements seem to be contradictory.

Furthermore, I believe that the direction of the association between neuroticism and effort regulation (e.g. Smillie et al., 2006) could also be the opposite of the authors’ interpretation e.g. high neuroticism lowers effort. There may well be arguments for the relationship to hold in either direction but some acknowledgement of the mixed evidence is required.

There is a literature on the relationship between listening effort (not reported in the current study unless DTT performance counts as listening effort) and personality. I don’t think the existing literature supports a relationship between neuroticism and “listening effort” (performance on a digits task) e.g. Bakan (1959). Interestingly, the authors report results that are inconsistent in some of their own earlier work (Tune et al. 2018) where no significant correlations between personality traits and performance on a digits task were identified.

The authors explain on Page 17 that questions about listening effort were asked but the data were not included in the submitted manuscript. These results could inform and strengthen the arguments presented in the Discussion.

2) Number of statistical comparisons

The authors have conducted many statistical tests but they also have a very large sample size. Were the results of the regression analyses reported on pages 6-7 corrected for number of regression models used, including the joint extraversion-neuroticism predictor? The predictors used in the various regression models are related.

3) Potential clinical applications

The authors frame their results in terms of potential to provide “...tailored audiological treatments...” (e.g. Abstract & Discussion). How will these results be used to tailor treatment in clinical practice? Should audiologists administer the BFI-S questionnaire?

The authors report statistical effect sizes but how do results relate to a clinically meaningful change in performance on the ANL, SSQ or DTT? Please could the authors expand on their argument that “...what appears to be a relatively small effect can amount nevertheless to considerable real-life relevance...” (Discussion, page 12). How does personality affect hearing/listening/speech tests over time?

Other comments

Abstract

The abstract needs some editing.

Line 15. "...background noise..." insert "levels" after.

Line 16. "...speech-in-noise comprehension..." insert "scores" after.

Line 18. "...the bias to subjectively..." suggest rephrase to "...the bias in subjectively over-rating..".

Lines 20-21. "Of benefit to...". Suggest this is rephrased. What is the "solid framework" that is referred to? The present results?

Line 25. "and health hazard" change to "and a health hazard".

Introduction

Page 4, Line 13. Remove "in" from "...unclear in how...".

Page 4, Line 16. "...solid framework on..." should read "...solid framework based on..." (?).

Page 4, Lines 23-24. "increased seeking for privacy..." suggest rephrase.

Page 4, Line 28. "it's opposite" suggest "it's counterpart".

Page 4, Line 28. "factor to understand" change to "factor in understanding".

Page 4, Line 33. "It is long known" suggest change to "It has long been known".

Page 4, Line 44. Control of potential confounders the relationship between personality and hearing/listening tests. Did you screen for tinnitus, hyperacusis?

Page 5, Line 8-9. "Hearing loss generally receives..." this sentence does not seem to belong here.

Page 5, Line 15. "...the limited benefit from hearing aids..." specifics are needed e.g. speech perception in noise.

Page 5 (and elsewhere). I welcome the use of pre-registered hypotheses but readers may benefit from a brief explanation of why certain personality traits may be more/less associated with noise tolerance (page 5).

Results

Page 6. The legend for Fig. 1 states that the results were obtained for "four established noise tolerance tests". The 4 outcome measures are not measures of "noise tolerance".

Page 7. DTT is described as a measure of speech-in-noise comprehension. Is this intentional?

Page 8. "...reveal a striking dissociation...". This dissociation was seen for neuroticism only and the effect size for DTT association seems weak.

Discussion

Page 11. How were other personality traits "rigorously controlled"?

Page 11. "pervasively used noise tolerance tests used in audiological...". Are the ANL, SSQ speech scale and DTT used in clinic? Perhaps this would vary from country to country according to national guidance.

Page 12. Suggest "she" changed to gender-neutral pronoun.

Methods

Page 17. Were the sample sizes determined a priori?

Review form: Reviewer 3

Is the manuscript scientifically sound in its present form?

Yes

Are the interpretations and conclusions justified by the results?

No

Is the language acceptable?

Yes

Do you have any ethical concerns with this paper?

No

Have you any concerns about statistical analyses in this paper?

No

Recommendation?

Major revision is needed (please make suggestions in comments)

Comments to the Author(s)

RSOS-210881

Personality captures dissociations of subjective versus objective noise tolerance

Wöstmann, Erb, Kreitewolf, Obleser

This study examined the relationship between personality and noise tolerance in a large sample of adults. More specifically, it evaluated the predictive value of extraversion and neuroticism scores from a brief Big Five inventory measure to subjective noise tolerance (self-report measures of noise resistance, speech in noise comprehension, and acceptable noise level) and to what the authors refer to as objective noise tolerance (performance on the digit triplet test (DTT)). The main findings are that, consistent with previous studies, higher levels of neuroticism and lower levels of extraversion are associated with lower subjective noise tolerance. Additionally, they found that higher neuroticism predicted better performance on the speech in noise task. They conclude that these findings suggest that personality captures a dissociation of subjective and objective noise tolerance. The authors go on to explore how neuroticism and extraversion and joint personality predictors describe individuals' over/under rating of their performance on the DTT. They make a case for this being an important distinction that could have implications for clinical audiology in the form of treatment considerations and recommendations.

Overall, the paper is well written and enjoyable to read. An examination of non-auditory factors that affect noise tolerance is interesting and the authors provide the reader with a sufficient background literature. However, I have major concerns regarding the methods and subsequently the way the manuscript is framed, and the results are interpreted.

My greatest concerns are the use and description of the DTT as an "objective measure of noise tolerance" and the lack of an additional measure of hearing sensitivity. As the authors describe in the discussion section, the DTT is highly correlated with pure tone average, which indicates that it is more a measure of signal audibility, than speech comprehension, and to my knowledge, it has never been described as a measure of tolerance of noise. I ask that the authors please provide more support for using the DTT and for characterizing it as a measure of noise tolerance. Additionally, I ask that they also address why a measure of hearing sensitivity was not included, to control for hearing loss.

My concern is that without accounting for hearing sensitivity in a way other than self-report, which is not highly correlated with behaviorally assessed hearing, it is not accurate to describe the results as showing a dissociation between subjective v. objective noise tolerance, but rather of different associations of personality with self-reported noise tolerance vs. speech perception in noise or hearing sensitivity. That the DTT is capturing signal audibility and hearing loss can be seen in this data set, as the greatest effects on performance were the speaker/headphone the participant used (those who used headphones performed better than those who completed the task in the sound field (internal or external speakers)) and how old they

were (older individuals performed much worse and, importantly, are also much more likely to have hearing loss). Neuroticism score, by comparison, had only a very small effect.

When viewed this way, it is not surprising that there is a difference between subjective reports of noise tolerance, and someone's ability to recall speech in background noise or their overall hearing ability, and that these reports and abilities show different relationships with personality traits. Clarifying the language seems especially important given the small effects reported in the paper. An alternative interpretation could be that neuroticism and extraversion predict how well someone feels they do in noise or can deal with noise (which has already been established in the literature), but that overall hearing ability or ability to understand words in background noise have a less well-defined relationship with personality.

In lieu of rerunning the study and including an additional hearing screening measure, to allow the authors to parse out the role of hearing sensitivity on the self-report measures and DTT performance, the authors could consider reframing the study. One possibility is to replace the term "objective measure of noise tolerance" with the standard language used for the test, speech perception in noise. The authors may consider using the language they used for their OSF folder, assessing how personality traits predict noise sensitivity and speech in noise comprehension – though the DTT is not necessarily a measure of comprehension, as previously discussed. This is a more conservative, and in this reviewer's opinion, more precise description of what was assessed under the present study conditions.

Additional points:

Table 1 seems to suggest that several other personality traits contribute to self-reported noise tolerance on a similar scale as neuroticism and extraversion. Can the authors comment on why they did not include some thoughts on these results in their discussion? It could add to the literature to more fully characterize the contribution of personality to self-reported noise tolerance.

Listening effort: The authors state in the methods that they asked questions about listening effort in the online study and they also bring listening effort into their discussion section, however they do not report on those findings in the present paper. Can the authors comment on why they did not include this variable in the current manuscript? This could be important, if those results support their point that the allocation of more cognitive resources for those with higher neuroticism could be one way to explain their findings of the dissociation between objective and subjective noise tolerance.

The common language effect sizes are all very close to 50%, so while it is appreciated that they used CL effect sizes, it does little to assuage my feelings that the effects are very small, although the authors do not generally describe them as such. Can the authors please discuss the size of the different effects they identified and provide confidence intervals for these effects?

Decision letter (RSOS-210881.R0)

Dear Dr Wöstmann

The Editors assigned to your paper RSOS-210881 "Personality captures dissociations of subjective versus objective noise tolerance" have now received comments from reviewers and would like

you to revise the paper in accordance with the reviewer comments and any comments from the Editors. Please note this decision does not guarantee eventual acceptance.

Please submit your revised manuscript and required files (see below) no later than 21 days from today's (ie 02-Aug-2021) date. Note: the ScholarOne system will 'lock' if submission of the revision is attempted 21 or more days after the deadline. If you do not think you will be able to meet this deadline please contact the editorial office immediately.

on behalf of Dr César Lima (Associate Editor) and Essi Viding (Subject Editor)
openscience@royalsociety.org

Reviewer comments to Author:

Reviewer: 1
Comments to the Author(s)

I think this is an interesting and useful study that seems to have been conducted quite well to the extent that I am able to evaluate it without being familiar with the statistical methods employed here. Despite my lack of experience with these methods, I think they are sufficiently well described for an interested and determined reader to follow up on adequately. Most importantly, the methods and results are presented with sufficient clarity to permit replication (and the availability of the preregistration, data, and analysis scripts is very helpful in this regard). I think the overall results contribute to our understanding of individual differences in noise sensitivity, and the exploratory analyses provide a basis for some intriguing avenues for future research. I have only a few suggestions for small changes that I think will improve the readability of the manuscript. (note - page numbers are based on those assigned by the editorial manager software, not the ones provided by the authors in the ms)

p. 17

around line 20 – Based on data from OSF it looks like school qualifications were from the German school system. Were participants all from Germany? From German-speaking regions? How was this ensured? A great deal of emphasis is put on the rigor of demographic selection, but not much is described in the methods.

Around line 33 – it seems as if the final analyses could have been done on either the complete sample N_{total} with missing values compensated for somehow, or on the sample that completed all tasks ($N_{total}-61$) but it's not clear which was the decision. As this is explained in a little more detail on p. 19, maybe it makes sense to just report the overall total here, and then go into the full detail in the data pre-processing section (especially since I don't see the $N=1042$ group in the data pre-processing explanation anyway).

around line 35 – I would recommend listing the 5 tasks here. It's a little redundant, but it will help readers who have skipped straight to the methods (as one does).

p. 18

around line 3 – I think it would be clearer to say that the ratio scores were then subtracted from 1, rather than “inverted” If I am correct that the score is $1 - (\text{score}/(21*6))$?

Around line 12 – please give detailed methods as in the Noise resistance score (to make it easy to replicate accurately). Which six items? What were the max scores for each item, etc.

Around line 21 – was there any way to ensure that participants were in fact adjusting the levels as requested? If not, how was the reliability of these measures ensured? Was there anything like the “catch trials” in the surveys?

Around line 38 – why were ANL scores not normalized to between 0 and 1 as the others were? Would having these scores be on a much larger and less-fine-grained scale affect the statistical analyses (i.e. the other variables in the mix range from 0 to 1 in 0.01 unit steps, while ANL scores seem to vary from -26 to 8 in steps of 2). (DTT ranges from -1.49 to 0.17 but there are only 31 unique values).

P 20

Around line 7 – Given that all subjects with missing data were excluded from that analysis (but, I think, not from other analyses for which they had given responses) I think it makes sense to report numbers of subjects as I described in the comment re. p. 17 – and I see that this was done nicely in Figure 2.

P 10

Around lines 50-55 I find the sentence “That is, higher scores on extraversion versus neuroticism were accompanied by overrated versus underrated own noise tolerance, respectively” to be very confusing. I would recommend unpacking it into multiple sentences, as this seems to be a crucial finding and one that might easily be missed or misunderstood from this phrasing.

P 12

Line 33 – “extravert” should be “extraverted”

Reviewer: 2

Comments to the Author(s)

RSOS-210881

The authors conducted a large scale study of the relationship(s) between personality traits and 4 measures of hearing/listening [Weinstein Noise Sensitivity Scale (WNSS), the Acceptable Noise

Level (ANL) test, the Speech items of the Speech, Spatial and Qualities (SSQ) scale and the Digits-Triplet-Test (DTT)]. The results suggest that different personality traits (neuroticism, and to a lesser extent extraversion) can impact results from hearing/listening tests.

I have some concerns over the interpretation of results (please see below).

1) Interpretation

1a) Outcomes described as measures of “noise tolerance”

I understand why the authors would want to simplify their findings and categorize all 4 outcome measures as measures of “noise tolerance”. I agree with the authors that 2 of the tests (Weinstein Noise Sensitivity Scale (WNSS) and the Acceptable Noise Level (ANL)) are tests of noise tolerance. But I am not clear on why the authors also describe the Speech items of the Speech, Spatial and Qualities (SSQ) scale and the Digits-Triplet-Test (DTT) as “noise tolerance tests”.

The authors describe the selected items of the SSQ (“Speech-hearing items”; Gatehouse and Noble, 2004) as a measure of “speech-in-noise comprehension”. Gatehouse & Noble used the term “speech-hearing ability” to describe what the SSQ measures.

The DTT is not a noise tolerance test. It can be used as a test of hearing acuity or screen for hearing loss e.g. the World Health Organisation’s hearWHO app. Indeed, on page 12 the authors state that the DTT primarily assesses hearing sensitivity.

The authors should revisit their descriptions of the SSQ and the DTT as “noise tolerance tests” (in the title and throughout).

1b) Interpretation re listening effort

On page 12 the authors argue that non-auditory factors such as memory or attention have a negligible impact on DTT performance. Then on page 13 the authors argue that people who are higher in neuroticism allocate more cognitive resource which “eventually results in lower speech reception thresholds in the DTT”. These statements seem to be contradictory.

Furthermore, I believe that the direction of the association between neuroticism and effort regulation (e.g. Smillie et al., 2006) could also be the opposite of the authors’ interpretation e.g. high neuroticism lowers effort. There may well be arguments for the relationship to hold in either direction but some acknowledgement of the mixed evidence is required.

There is a literature on the relationship between listening effort (not reported in the current study unless DTT performance counts as listening effort) and personality. I don’t think the existing literature supports a relationship between neuroticism and “listening effort” (performance on a digits task) e.g. Bakan (1959). Interestingly, the authors report results that are inconsistent in some of their own earlier work (Tune et al. 2018) where no significant correlations between personality traits and performance on a digits task were identified.

The authors explain on Page 17 that questions about listening effort were asked but the data were not included in the submitted manuscript. These results could inform and strengthen the arguments presented in the Discussion.

2) Number of statistical comparisons

The authors have conducted many statistical tests but they also have a very large sample size. Were the results of the regression analyses reported on pages 6-7 corrected for number of

regression models used, including the joint extraversion-neuroticism predictor? The predictors used in the various regression models are related.

3) Potential clinical applications

The authors frame their results in terms of potential to provide "...tailored audiological treatments..." (e.g. Abstract & Discussion). How will these results be used to tailor treatment in clinical practice? Should audiologists administer the BFI-S questionnaire?

The authors report statistical effect sizes but how do results relate to a clinically meaningful change in performance on the ANL, SSQ or DTT? Please could the authors expand on their argument that "...what appears to be a relatively small effect can amount nevertheless to considerable real-life relevance..." (Discussion, page 12). How does personality affect hearing/listening/speech tests over time?

Other comments

Abstract

The abstract needs some editing.

Line 15. "...background noise..." insert "levels" after.

Line 16. "...speech-in-noise comprehension..." insert "scores" after.

Line 18. "...the bias to subjectively..." suggest rephrase to "...the bias in subjectively over-rating..".

Lines 20-21. "Of benefit to...". Suggest this is rephrased. What is the "solid framework" that is referred to? The present results?

Line 25. "and health hazard" change to "and a health hazard".

Introduction

Page 4, Line 13. Remove "in" from "...unclear in how...".

Page 4, Line 16. "...solid framework on..." should read "...solid framework based on..." (?).

Page 4, Lines 23-24. "increased seeking for privacy..." suggest rephrase.

Page 4, Line 28. "it's opposite" suggest "it's counterpart".

Page 4, Line 28. "factor to understand" change to "factor in understanding".

Page 4, Line 33. "It is long known" suggest change to "It has long been known".

Page 4, Line 44. Control of potential confounders the relationship between personality and hearing/listening tests. Did you screen for tinnitus, hyperacusis?

Page 5, Line 8-9. "Hearing loss generally receives..." this sentence does not seem to belong here.

Page 5, Line 15. "...the limited benefit from hearing aids..." specifics are needed e.g. speech perception in noise.

Page 5 (and elsewhere). I welcome the use of pre-registered hypotheses but readers may benefit from a brief explanation of why certain personality traits may be more/less associated with noise tolerance (page 5).

Results

Page 6. The legend for Fig. 1 states that the results were obtained for "four established noise tolerance tests". The 4 outcome measures are not measures of "noise tolerance".

Page 7. DTT is described as a measure of speech-in-noise comprehension. Is this intentional?

Page 8. "...reveal a striking dissociation...". This dissociation was seen for neuroticism only and the effect size for DTT association seems weak.

Discussion

Page 11. How were other personality traits "rigorously controlled"?

Page 11. “pervasively used noise tolerance tests used in audiological...”. Are the ANL, SSQ speech scale and DTT used in clinic? Perhaps this would vary from country to country according to national guidance.

Page 12. Suggest “she” changed to gender-neutral pronoun.

Methods

Page 17. Were the sample sizes determined a priori?

Reviewer: 3

Comments to the Author(s)

RSOS-210881

Personality captures dissociations of subjective versus objective noise tolerance

Wöstmann, Erb, Kreitewolf, Obleser

This study examined the relationship between personality and noise tolerance in a large sample of adults. More specifically, it evaluated the predictive value of extraversion and neuroticism scores from a brief Big Five inventory measure to subjective noise tolerance (self-report measures of noise resistance, speech in noise comprehension, and acceptable noise level) and to what the authors refer to as objective noise tolerance (performance on the digit triplet test (DTT)). The main findings are that, consistent with previous studies, higher levels of neuroticism and lower levels of extraversion are associated with lower subjective noise tolerance. Additionally, they found that higher neuroticism predicted better performance on the speech in noise task. They conclude that these findings suggest that personality captures a dissociation of subjective and objective noise tolerance. The authors go on to explore how neuroticism and extraversion and joint personality predictors describe individuals' over/under rating of their performance on the DTT. They make a case for this being an important distinction that could have implications for clinical audiology in the form of treatment considerations and recommendations.

Overall, the paper is well written and enjoyable to read. An examination of non-auditory factors that affect noise tolerance is interesting and the authors provide the reader with a sufficient background literature. However, I have major concerns regarding the methods and subsequently the way the manuscript is framed, and the results are interpreted.

My greatest concerns are the use and description of the DTT as an “objective measure of noise tolerance” and the lack of an additional measure of hearing sensitivity. As the authors describe in the discussion section, the DTT is highly correlated with pure tone average, which indicates that it is more a measure of signal audibility, than speech comprehension, and to my knowledge, it has never been described as a measure of tolerance of noise. I ask that the authors please provide more support for using the DTT and for characterizing it as a measure of noise tolerance.

Additionally, I ask that they also address why a measure of hearing sensitivity was not included, to control for hearing loss.

My concern is that without accounting for hearing sensitivity in a way other than self-report, which is not highly correlated with behaviorally assessed hearing, it is not accurate to describe the results as showing a dissociation between subjective v. objective noise tolerance, but rather of different associations of personality with self-reported noise tolerance vs. speech perception in noise or hearing sensitivity. That the DTT is capturing signal audibility and hearing loss can be seen in this data set, as the greatest effects on performance were the speaker/headphone the participant used (those who used headphones performed better than those who completed the task in the sound field (internal or external speakers)) and how old they were (older individuals performed much worse and, importantly, are also much more likely to have hearing loss).

Neuroticism score, by comparison, had only a very small effect.

When viewed this way, it is not surprising that there is a difference between subjective reports of noise tolerance, and someone's ability to recall speech in background noise or their overall hearing ability, and that these reports and abilities show different relationships with personality

traits. Clarifying the language seems especially important given the small effects reported in the paper. An alternative interpretation could be that neuroticism and extraversion predict how well someone feels they do in noise or can deal with noise (which has already been established in the literature), but that overall hearing ability or ability to understand words in background noise have a less well-defined relationship with personality.

In lieu of rerunning the study and including an additional hearing screening measure, to allow the authors to parse out the role of hearing sensitivity on the self-report measures and DTT performance, the authors could consider reframing the study. One possibility is to replace the term “objective measure of noise tolerance” with the standard language used for the test, speech perception in noise. The authors may consider using the language they used for their OSF folder, assessing how personality traits predict noise sensitivity and speech in noise comprehension – though the DTT is not necessarily a measure of comprehension, as previously discussed. This is a more conservative, and in this reviewer’s opinion, more precise description of what was assessed under the present study conditions.

Additional points:

Table 1 seems to suggest that several other personality traits contribute to self-reported noise tolerance on a similar scale as neuroticism and extraversion. Can the authors comment on why they did not include some thoughts on these results in their discussion? It could add to the literature to more fully characterize the contribution of personality to self-reported noise tolerance.

Listening effort: The authors state in the methods that they asked questions about listening effort in the online study and they also bring listening effort into their discussion section, however they do not report on those findings in the present paper. Can the authors comment on why they did not include this variable in the current manuscript? This could be important, if those results support their point that the allocation of more cognitive resources for those with higher neuroticism could be one way to explain their findings of the dissociation between objective and subjective noise tolerance.

The common language effect sizes are all very close to 50%, so while it is appreciated that they used CL effect sizes, it does little to assuage my feelings that the effects are very small, although the authors do not generally describe them as such. Can the authors please discuss the size of the different effects they identified and provide confidence intervals for these effects?

===PREPARING YOUR MANUSCRIPT===

===PREPARING YOUR REVISION IN SCHOLARONE===

<https://royalsociety.org/journals/authors/author-guidelines/#data>. You should ensure that

you cite the dataset in your reference list. If you have deposited data etc in the Dryad repository, please include both the 'For publication' link and 'For review' link at this stage.

Author's Response to Decision Letter for (RSOS-210881.R0)

See Appendix A.

RSOS-210881.R0 (Original submission)

Review form: Reviewer 2

Is the manuscript scientifically sound in its present form?

Yes

Are the interpretations and conclusions justified by the results?

Yes

Is the language acceptable?

Yes

Do you have any ethical concerns with this paper?

No

Have you any concerns about statistical analyses in this paper?

No

Recommendation?

Accept as is

Comments to the Author(s)

The authors have been very responsive and thorough in addressing my comments. I thank them for the time and effort they have put into revising their manuscript.

Review form: Reviewer 3

Is the manuscript scientifically sound in its present form?

Yes

Are the interpretations and conclusions justified by the results?

Yes

Is the language acceptable?

Yes

Do you have any ethical concerns with this paper?

No

Have you any concerns about statistical analyses in this paper?

No

Recommendation?

Accept with minor revision (please list in comments)

Comments to the Author(s)

RSOS-210881.R1

Personality captures dissociations of subjective versus objective hearing in noise
Wöstmann, Erb, Kreitewolf, Obleser

I thank the authors for their careful consideration of my comments on the initial submission. It is clear that they took all reviewer comments and concerns under advisement and made a considerable number of modifications, which have strengthened the manuscript. My major concerns have been adequately addressed. Below are some minor concerns for the authors to consider.

- P17 Ln 27: This reference is quite old and is likely not reflective of the current relationship between noise sensitivity and hearing aid use, given the considerable advances in hearing aids in the last 30 years (e.g. digital vs. analog systems, noise reduction algorithms, directional microphones, etc). The authors may consider removing this statement unless a more current paper supports this sentiment.
- P18 Ln 8-23: This paragraph is a bit hard to follow. I recommend adding “of hearing loss” to the sentence on line 13, which will help that sentence and the one following it. Additionally, as written, line 17 implies that hearing aids have limited benefit for speech perception in noise. This is true for some hearing aid users, and likely affects their success/satisfaction with the devices but hearing aids DO generally improve speech perception in background noise. I believe the authors are trying to describe that for individuals who report limited hearing aid benefit in noise, audiologic and demographic factors do not fully describe this lack of benefit. Please consider revising, for clarity.
- The authors may consider including the results of -Nabelek, Tampas, Burchfield (2004). Comparison of speech perception in background noise with acceptance of background noise in aided and unaided conditions. JSLHR. 47:1001-1011- in the background, as it directly relates to their goal of understanding subjective vs. objective hearing in noise.
- Throughout the document there are instances where the authors describe hearing in noise ability as high/low, which I felt detracted from the readability. It is likely that this is

partially due to switching out “noise tolerance” with “hearing in noise” and in some cases it doesn’t quite work with the same wording. Below is a list of the instances that I found to be problematic/unclear and which the authors should consider modifying to include language to help the reader (e.g. worse/poorer instead of lower, better instead of higher)

P 17, Ln 57

P 19 Ln 10-11

P 23 Ln 3, 7

P 26, Ln 27

P 27, Ln 33

Decision letter (RSOS-210881.R1)

Dear Dr Wöstmann

On behalf of the Editors, we are pleased to inform you that your Manuscript RSOS-210881.R1 "Personality captures dissociations of subjective versus objective hearing in noise" has been accepted for publication in Royal Society Open Science subject to minor revision in accordance with the referees' reports. Please find the referees' comments along with any feedback from the Editors below my signature.

Please submit your revised manuscript and required files (see below) no later than 7 days from today's (ie 06-Oct-2021) date. Note: the ScholarOne system will 'lock' if submission of the revision is attempted 7 or more days after the deadline. If you do not think you will be able to meet this deadline please contact the editorial office immediately.

on behalf of Dr César Lima (Associate Editor) and Essi Viding (Subject Editor)
openscience@royalsociety.org

Reviewer comments to Author:

Reviewer: 2

Comments to the Author(s)

The authors have been very responsive and thorough in addressing my comments. I thank them for the time and effort they have put into revising their manuscript.

Reviewer: 3

Comments to the Author(s)

RSOS-210881.R1

Personality captures dissociations of subjective versus objective hearing in noise
Wöstmann, Erb, Kreitewolf, Obleser

I thank the authors for their careful consideration of my comments on the initial submission. It is clear that they took all reviewer comments and concerns under advisement and made a considerable number of modifications, which have strengthened the manuscript. My major concerns have been adequately addressed. Below are some minor concerns for the authors to consider.

- P17 Ln 27: This reference is quite old and is likely not reflective of the current relationship between noise sensitivity and hearing aid use, given the considerable advances in hearing aids in the last 30 years (e.g. digital vs. analog systems, noise reduction algorithms, directional microphones, etc). The authors may consider removing this statement unless a more current paper supports this sentiment.
- P18 Ln 8-23: This paragraph is a bit hard to follow. I recommend adding “of hearing loss” to the sentence on line 13, which will help that sentence and the one following it. Additionally, as written, line 17 implies that hearing aids have limited benefit for speech perception in noise. This is true for some hearing aid users, and likely affects their success/satisfaction with the devices but hearing aids DO generally improve speech perception in background noise. I believe the authors are trying to describe that for individuals who report limited hearing aid benefit in noise, audiologic and demographic factors do not fully describe this lack of benefit. Please consider revising, for clarity.
- The authors may consider including the results of -Nabelek, Tampas, Burchfield (2004). Comparison of speech perception in background noise with acceptance of background noise in aided and unaided conditions. JSLHR. 47:1001-1011- in the background, as it directly relates to their goal of understanding subjective vs. objective hearing in noise.
- Throughout the document there are instances where the authors describe hearing in noise ability as high/low, which I felt detracted from the readability. It is likely that this is partially due to switching out “noise tolerance” with “hearing in noise” and in some cases it doesn’t quite work with the same wording. Below is a list of the instances that I found to be problematic/unclear and which the authors should consider modifying to include language to help the reader (e.g. worse/poorer instead of lower, better instead of higher)
P 17, Ln 57
P 19 Ln 10-11
P 23 Ln 3, 7
P 26, Ln 27
P 27, Ln 33

===PREPARING YOUR MANUSCRIPT===

===PREPARING YOUR REVISION IN SCHOLARONE===

- An individual file of each figure (EPS or print-quality PDF preferred [either format should be produced directly from original creation package], or original software format).
- An editable file of each table (.doc, .docx, .xls, .xlsx, or .csv).
- An editable file of all figure and table captions.

- Any electronic supplementary material (ESM).
- If you are requesting a discretionary waiver for the article processing charge, the waiver form must be included at this step.
- If you are providing image files for potential cover images, please upload these at this step, and inform the editorial office you have done so. You must hold the copyright to any image provided.
- A copy of your point-by-point response to referees and Editors. This will expedite the preparation of your proof.

- Ensure that your data access statement meets the requirements at <https://royalsociety.org/journals/authors/author-guidelines/#data>. You should ensure that you cite the dataset in your reference list. If you have deposited data etc in the Dryad repository, please only include the 'For publication' link at this stage. You should remove the 'For review' link.
- If you are requesting an article processing charge waiver, you must select the relevant waiver option (if requesting a discretionary waiver, the form should have been uploaded at Step 3 'File upload' above).
- If you have uploaded ESM files, please ensure you follow the guidance at <https://royalsociety.org/journals/authors/author-guidelines/#supplementary-material> to include a suitable title and informative caption. An example of appropriate titling and captioning may be found at https://figshare.com/articles/Table_S2_from_Is_there_a_trade-off_between_peak_performance_and_performance_breadth_across_temperatures_for_aerobic_scope_in_teleost_fishes_/3843624.

Author's Response to Decision Letter for (RSOS-210881.R1)

See Appendix B.

Decision letter (RSOS-210881.R2)

Dear Dr Wöstmann,

I am pleased to inform you that your manuscript entitled "Personality captures dissociations of subjective versus objective hearing in noise" is now accepted for publication in Royal Society Open Science.

on behalf of Dr César Lima (Associate Editor) and Essi Viding (Subject Editor)
openscience@royalsociety.org

Department of Psychology, University of Lübeck, Maria-Goeppert Straße 9a, 23562 Lübeck, Germany

Lübeck, September 10, 2021

Dear Royal Society Open Science editorial staff,

we are grateful for the invitation to submit a revised version of our manuscript RSOS-210881.

To address the reviewers' comments, we added additional sections to our manuscript, implemented changes to the terminology, and report additional analysis on our measure of listening effort. We believe that the revision has improved the manuscript and we thank the reviewers for their thoughtful comments.

Our point-by-point reply to the reviewers' comments can be found below. In the revised manuscript, all changes are highlighted in blue font.

We would be most grateful should this thoroughly revised version be deemed suitable for publication in *Royal Society Open Science*. Also, on behalf of our co-authors, thank you in advance for your consideration and we look forward to hearing from you.

Sincerely,

Malte Wöstmann and Jonas Obleser

Reviewer: 1

Comments to the Author(s)

I think this is an interesting and useful study that seems to have been conducted quite well to the extent that I am able to evaluate it without being familiar with the statistical methods employed here. Despite my lack of experience with these methods, I think they are sufficiently well described for an interested and determined reader to follow up on adequately. Most importantly, the methods and results are presented with sufficient clarity to permit replication (and the availability of the preregistration, data, and analysis scripts is very helpful in this regard). I think the overall results contribute to our understanding of individual differences in noise sensitivity, and the exploratory analyses provide a basis for some intriguing avenues for future research. I have only a few suggestions for small changes that I think will improve the readability of the manuscript. (note - page numbers are based on those assigned by the editorial manager software, not the ones provided by the authors in the ms)

Authors: We thank the reviewer for the positive assessment of our manuscript.

p. 17

around line 20 – Based on data from OSF it looks like school qualifications were from the German school system. Were participants all from Germany? From German-speaking regions? How was this ensured? A great deal of emphasis is put on the rigor of demographic selection, but not much is described in the methods.

Authors: We appreciate this comment and address it in the revised Methods section (p. 17):

Participants were recruited via crowdsourcing. The study was only available to participants who, according to their registration with the crowdsourcing platform, were German.

Around line 33 – it seems as if the final analyses could have been done on either the complete sample N_{total} with missing values compensated for somehow, or on the sample that completed all tasks ($N_{total}-61$) but it's not clear which was the decision. As this is explained in a little more detail on p. 19, maybe it makes sense to just report the overall total here, and then go into the full detail in the data pre-processing section (especially since I don't see the $N=1042$ group in the data pre-processing explanation anyway).

around line 35 – I would recommend listing the 5 tasks here. It's a little redundant, but it will help readers who have skipped straight to the methods (as one does).

Authors: In line with the reviewer's suggestions, we only report the overall N in the respective section since all further details are reported in the Data pre-processing section. We also list the five tasks to enhance readability.

p. 18

around line 3 – I think it would be clearer to say that the ratio scores were then subtracted from 1, rather than "inverted" If I am correct that the score is $1 - (\text{score}/(21*6))$?

Authors: In line with the reviewer's suggestion, we replaced "inverted" by "subtracted from 1".

Around line 12 – please give detailed methods as in the Noise resistance score (to make it easy to replicate accurately). Which six items? What were the max scores for each item, etc.

Authors: In the revised manuscript, we added this missing information (p. 18):

Six items of the Speech, Spatial and Qualities questionnaire (SSQ; items Speech 1, 2, 3, 6, 7, & 10 from Gatehouse & Noble, 2004; translated to German according to Kießling, Grugel, Meister, & Meis, 2011) were used to assess self-rated speech-hearing ability under challenging conditions. The range of possible scores (0–10) was scaled between 0 and 1, with higher scores reflecting better speech-hearing ability.

Around line 21 – was there any way to ensure that participants were in fact adjusting the levels as requested? If not, how was the reliability of these measures ensured? Was there anything like the "catch trials" in the surveys?

Authors: Instead of using "catch trials", reliability of ANL scores was ensured by only accepting ANL scores in-between SNL and LNL scores as valid. We describe this procedure in the Methods section (p. 19):

The strictest exclusion criteria were applied to the ANL test, where an acceptable noise level (ANL) was only considered valid if it was in-between the soft noise levels (SNL) and the loud noise level (LNL) and removed otherwise.

Around line 38 – why were ANL scores not normalized to between 0 and 1 as the others were? Would having these scores be on a much larger and less-fine-grained scale affect the statistical analyses (i.e. the other variables in the mix range from 0 to 1 in 0.01 unit steps, while ANL scores seem to vary from -26 to 8 in steps of 2). (DTT ranges from -1.49 to 0.17 but there are only 31 unique values).

Authors: Our rationale was to scale results of questionnaires (BFI-S, SSQ, WNSS), which all used somewhat different scales, between 0 and 1. In an additional control analysis for review only, we also scaled ANL and DTT between 0 and 1, which did not have any effect on the results.

P 20

Around line 7 – Given that all subjects with missing data were excluded from that analysis (but, I think, not from other analyses for which they had given responses) I think it makes sense to report numbers of subjects as I described in the comment re. p. 17 – and I see that this was done nicely in Figure 2.

Authors: In line with the reviewer's recommendation, we now mention the exact number of participants for each regression analysis also in the Methods section.

P 10

Around lines 50-55 I find the sentence "That is, higher scores on extraversion versus neuroticism were accompanied by overrated versus underrated own noise tolerance, respectively" to be very confusing.

I would recommend unpacking it into multiple sentences, as this seems to be a crucial finding and one that might easily be missed or misunderstood from this phrasing.

P 12

Line 33 – “extravert” should be “extraverted”

Authors: In the revised manuscript, we unpacked this sentence to enhance readability and corrected the typo.

Reviewer: 2

Comments to the Author(s)

RSOS-210881

The authors conducted a large scale study of the relationship(s) between personality traits and 4 measures of hearing/listening [Weinstein Noise Sensitivity Scale (WNSS), the Acceptable Noise Level (ANL) test, the Speech items of the Speech, Spatial and Qualities (SSQ) scale and the Digits-Triplet-Test (DTT)]. The results suggest that different personality traits (neuroticism, and to a lesser extent extraversion) can impact results from hearing/listening tests.

I have some concerns over the interpretation of results (please see below).

Authors: We thank the reviewer for the assessment of our manuscript.

1) Interpretation

1a) Outcomes described as measures of “noise tolerance”

I understand why the authors would want to simplify their findings and categorize all 4 outcome measures as measures of “noise tolerance”. I agree with the authors that 2 of the tests (Weinstein Noise Sensitivity Scale (WNSS) and the Acceptable Noise Level (ANL)) are tests of noise tolerance. But I am not clear on why the authors also describe the Speech items of the Speech, Spatial and Qualities (SSQ) scale and the Digits-Triplet-Test (DTT) as “noise tolerance tests”.

Authors: We agree with the reviewer that our use of the term “noise tolerance” does not adequately capture some of the tests used here. In the revised manuscript, we addressed this concern in several ways:

- We avoid the term “noise tolerance” in the title of the manuscript, which has been changed to: *Personality captures dissociations of subjective versus objective hearing in noise*
- Throughout the manuscript, we have made sure we use the term “noise tolerance” only when referring to WNSS and ANL, but the broader term “hearing in noise” when referring to SSQ and DTT or to the conglomerate of all four tests.
- We also changed the labels in Figure 3 to avoid the term “noise tolerance”.

The authors describe the selected items of the SSQ (“Speech-hearing items”; Gatehouse and Noble, 2004) as a measure of “speech-in-noise comprehension”. Gatehouse & Noble used the term “speech-hearing ability” to describe what the SSQ measures.

The DTT is not a noise tolerance test. It can be used as a test of hearing acuity or screen for hearing loss e.g. the World Health Organisation's hearWHO app. Indeed, on page 12 the authors state that the DTT primarily assesses hearing sensitivity.

The authors should revisit their descriptions of the SSQ and the DTT as "noise tolerance tests" (in the title and throughout).

Authors: In line with the reviewer's comments, we now refer to the SSQ as a measure of "speech-hearing ability" in text and figures. Furthermore, we avoid the term "noise tolerance" when referring to SSQ and DTT.

1b) Interpretation re listening effort

On page 12 the authors argue that non-auditory factors such as memory or attention have a negligible impact on DTT performance. Then on page 13 the authors argue that people who are higher in neuroticism allocate more cognitive resource which "eventually results in lower speech reception thresholds in the DTT". These statements seem to be contradictory.

Authors: We thank the reviewer for making us aware of this contradiction. Indeed, our statement that "factors such as working memory or attention are negligible" for predicting performance in the DTT test was somewhat misleading. Indeed, the study by Heinrich et al 2015 shows a relation of DTT scores and a measure of dual attention (Table S1; $p < .07$), which suggest some relation to attention (although considerably smaller than the relation to hearing thresholds). In the revised manuscript, we changed the respective section in the Discussion (p. 12):

Research suggests that the DTT primarily assesses hearing sensitivity (Armstrong, Oosterloo, Croll, Ikram, & Goedegebure, 2020; Koole et al., 2016) and that associations with non-auditory factors such as working memory or attention are considerably smaller (Heinrich, Henshaw, & Ferguson, 2015).

Furthermore, I believe that the direction of the association between neuroticism and effort regulation (e.g. Smillie et al., 2006) could also be the opposite of the authors' interpretation e.g. high neuroticism lowers effort. There may well be arguments for the relationship to hold in either direction but some acknowledgement of the mixed evidence is required.

There is a literature on the relationship between listening effort (not reported in the current study unless DTT performance counts as listening effort) and personality. I don't think the existing literature supports a relationship between neuroticism and "listening effort" (performance on a digits task) e.g. Bakan (1959). Interestingly, the authors report results that are inconsistent in some of their own earlier work (Tune et al. 2018) where no significant correlations between personality traits and performance on a digits task were identified.

Authors: We agree with the reviewer that in theory, a negative association of neuroticism and expended effort is also plausible. Indeed, the evidence available in the literature is mixed and we acknowledge this in the revised Discussion (p. 13&14):

Some positive associations were found for neuroticism and self-reported (Boyes & French, 2010), as well as physiological measures of effort (Mandell, Becker, VanAndel, Nelson, & Shaw, 2015). Of note, evidence for the direction of the association of neuroticism and effort is mixed. For instance, the

existing literature also includes studies that found no association of neuroticism and auditory vigilance (Bakan, 1959), as well as evidence that separate aspects of neuroticism might relate differently to task performance (van Doorn & Lang, 2010).

It is true that the studies by Bakan (1959) and Tune et al (2018) show non-significant associations between neuroticism and task performance. However, between-subject correlation/regression analyses with sample sizes of $N = 40$ (Bakan 1959) and $N = 29$ (Tune et al 2018) have very low power ($< 15\%$) to detect an association of neuroticism and task performance with a similar magnitude as in the present study. We hope that the reviewer agrees with us that we should not over-interpret these null-results.

Furthermore, we would like to mention that there is another study (Strand et al 2018, JSLHR), which supports the positive relation of neuroticism and listening effort (in a sample of $N > 100$). Although this study does not show these results in the published article, a significant positive relation between *Negative Emotionality* (a construct closely related to neuroticism) and the NASA-TASK LOAD index can be seen in online data of this study. These results are in line with the view of higher expended effort in individuals scoring higher on neuroticism.

The authors explain on Page 17 that questions about listening effort were asked but the data were not included in the submitted manuscript. These results could inform and strengthen the arguments presented in the Discussion.

Authors: We agree with the reviewer and now briefly refer to these data in the Discussion section. However, since we included only two questions (one asking for mental demand and the other for expended effort in the DTT), we cannot formally evaluate reliability of these measures and we are thus careful to not over-interpret the results. The following section has been added to the revised Discussion (p. 14):

In the second cohort of the present study, we included two questions following the DTT to probe participants' experienced mental demand and expended effort. Under statistical control for potential confounders (age, gender, sound presentation, highest education, hearing loss), neuroticism was a significant positive predictor of expended effort ($\beta = 0.099$; $p = 0.031$) but not of experienced mental demand ($\beta = 0.077$; $p = 0.091$). These results are consistent with the view of higher expended listening effort being associated with higher levels of neuroticism.

In the revised Methods section, we briefly introduce the two effort questions used in cohort 2 (p. 19):

After the DTT test, participants in cohort two completed two short questionnaires to assess some aspects related to listening effort. Two questions (adapted from the NASA-task load index; Hart, 2006) were intended to assess mental demand ("How high was your mental demand for this listening test?"; translated from German) and expended effort ("How hard did you have to work to reach your level of performance?"; translated from German) on a 10-point scale (1, very low; 10, very high).

2) Number of statistical comparisons

The authors have conducted many statistical tests but they also have a very large sample size. Were the results of the regression analyses reported on pages 6-7 corrected for number of regression models used, including the joint extraversion-neuroticism predictor? The predictors used in the various regression models are related.

Authors: We thank the reviewer for making us aware that this information was missing in the previous version of the manuscript. We would like to emphasize here that the major pre-registered analysis included 4 multiple regression models (Figure 2). To control for false positives, we did not correct the p-values for the number of tests but instead tested the replicability of individual effects by comparing all effects for the first and second cohort of participants. In the revised Methods section, we added the following information (p. 20):

The major statistical analysis comprised four multiple regression models (Fig. 2). To control for false positives, we did not correct p-values for the number of performed tests but instead tested the replicability of all effects by splitting up the dataset into the two cohorts. Bivariate Pearson correlations among predictors in regression models were small to moderate (all $|r| < .35$; see also Fig. 1b) and the Variance Inflation Factors of regression models were small (all $VIF < 2$), such that multicollinearity was not an issue.

3) Potential clinical applications

The authors frame their results in terms of potential to provide "...tailored audiological treatments..." (e.g. Abstract & Discussion). How will these results be used to tailor treatment in clinical practice? Should audiologists administer the BFI-S questionnaire?

The authors report statistical effect sizes but how do results relate to a clinically meaningful change in performance on the ANL, SSQ or DTT? Please could the authors expand on their argument that "...what appears to be a relatively small effect can amount nevertheless to considerable real-life relevance..." (Discussion, page 12). How does personality affect hearing/listening/speech tests over time?

Authors: In the end of the revised Discussion, we elaborate on these points in more detail (p. 15&16):

In practice, short personality inventories might be used to obtain knowledge about the likelihood of a specific individual to under- or overestimate their own hearing in noise, relative to other individuals. This could pose valuable clinical information and help manage client expectations in the extended period of fitting a new hearing device to an individual. For instance, if an individual scores particularly high on neuroticism, it is worth considering the possibility that this individual might tend to underrate hearing in noise abilities. This might also help to explain potential dissatisfaction with a hearing device.

However, it must be noted that although small associations of personality and hearing in noise likely cumulate over time in real-life, it should not be expected that audiologists detect an obvious association of personality and a single audiological screening test on the level of the individual client. At the present stage, the most important conclusion of this study for clinical audiology is that audiologists and their clients should be sensitized to the fact that subjective and objective measures of hearing in noise often dissociate, and that personality explains part of this dissociation.

Other comments

Abstract

The abstract needs some editing.

Line 15. "...background noise..." insert "levels" after.

Line 16. "...speech-in-noise comprehension..." insert "scores" after.

Line 18. "...the bias to subjectively..." suggest rephrase to "...the bias in subjectively over-rating..".

Lines 20-21. "Of benefit to...". Suggest this is rephrased. What is the "solid framework" that is referred to? The present results?

Line 25. "and health hazard" change to "and a health hazard".

Authors: We addressed all of these points in line with the reviewer's suggestions.

Introduction

Page 4, Line 13. Remove "in" from "...unclear in how...".

Page 4, Line 16. "...solid framework on..." should read "...solid framework based on..." (?).

Page 4, Lines 23-24. "increased seeking for privacy..." suggest rephrase.

Page 4, Line 28. "it's opposite" suggest "it's counterpart".

Page 4, Line 28. "factor to understand" change to "factor in understanding".

Page 4, Line 33. "It is long known" suggest change to "It has long been known".

Page 4, Line 44. Control of potential confounders the relationship between personality and hearing/listening tests. Did you screen for tinnitus, hyperacusis?

Page 5, Line 8-9. "Hearing loss generally receives..." this sentence does not seem to belong here.

Page 5, Line 15. "...the limited benefit from hearing aids..." specifics are needed e.g. speech perception in noise.

Page 5 (and elsewhere). I welcome the use of pre-registered hypotheses but readers may benefit from a brief explanation of why certain personality traits may be more/less associated with noise tolerance (page 5).

Authors: We addressed all of these comments. We did not screen for tinnitus and hyperacusis here but we will consider doing so in upcoming studies.

Results

Page 6. The legend for Fig. 1 states that the results were obtained for "four established noise tolerance tests". The 4 outcome measures are not measures of "noise tolerance".

Page 7. DTT is described as a measure of speech-in-noise comprehension. Is this intentional?

Page 8. "...reveal a striking dissociation...". This dissociation was seen for neuroticism only and the effect size for DTT association seems weak.

Authors: We addressed all these comments. We now refer to the DTT test as a test of speech-in-noise recognition or reception, in agreement with Smits et al (2013) JASA.

Discussion

Page 11. How were other personality traits "rigorously controlled"?

Page 11. "pervasively used noise tolerance tests used in audiological...". Are the ANL, SSQ speech scale and DTT used in clinic? Perhaps this would vary from country to country according to national guidance.

Page 12. Suggest "she" changed to gender-neutral pronoun.

Authors: We addressed these comments in the revised manuscript. We emphasize that we employed statistical control for other personality traits.

Methods

Page 17. Were the sample sizes determined a priori?

Authors: We state the rationale for the sample size in more detail in the revised Methods (p. 17):

The planned sample size in cohort one was at least $N = 500$. To be able to replicate the obtained results, the planned sample size in cohort two was also at least $N = 500$.

Reviewer: 3

Comments to the Author(s)

RSOS-210881

Personality captures dissociations of subjective versus objective noise tolerance
Wöstmann, Erb, Kreitewolf, Obleser

This study examined the relationship between personality and noise tolerance in a large sample of adults. More specifically, it evaluated the predictive value of extraversion and neuroticism scores from a brief Big Five inventory measure to subjective noise tolerance (self-report measures of noise resistance, speech in noise comprehension, and acceptable noise level) and to what the authors refer to as objective noise tolerance (performance on the digit triplet test (DTT)). The main findings are that, consistent with previous studies, higher levels of neuroticism and lower levels of extraversion are associated with lower subjective noise tolerance. Additionally, they found that higher neuroticism predicted better performance on the speech in noise task. They conclude that these findings suggest that personality captures a dissociation of subjective and objective noise tolerance. The authors go on to explore how neuroticism and extraversion and joint personality predictors describe individuals' over/under rating of their performance on the DTT. They make a case for this being an important distinction that could have implications for clinical audiology in the form of treatment considerations and recommendations.

Overall, the paper is well written and enjoyable to read. An examination of non-auditory factors that affect noise tolerance is interesting and the authors provide the reader with a sufficient background literature. However, I have major concerns regarding the methods and subsequently the way the manuscript is framed, and the results are interpreted.

Authors: We are grateful for the reviewer's assessment of our manuscript.

My greatest concerns are the use and description of the DTT as an "objective measure of noise tolerance" and the lack of an additional measure of hearing sensitivity. As the authors describe in the discussion section, the DTT is highly correlated with pure tone average, which indicates that it is more

a measure of signal audibility, than speech comprehension, and to my knowledge, it has never been described as a measure of tolerance of noise. I ask that the authors please provide more support for using the DTT and for characterizing it as a measure of noise tolerance. Additionally, I ask that they also address why a measure of hearing sensitivity was not included, to control for hearing loss.

Authors: We agree with the reviewer that the term “noise tolerance” might not adequately capture what some of the tests used do measure (especially the DTT). Reviewer #2 raised a similar concern. In the revised manuscript, we addressed this concern in several ways:

- We avoid the term “noise tolerance” in the title of the manuscript, which has been changed to: *Personality captures dissociations of subjective versus objective hearing in noise*
- Throughout the manuscript, we have made sure we use the term “noise tolerance” only when referring to WNSS and ANL, but the broader term “hearing in noise” when referring to SSQ and DTT or to the conglomerate of all four tests.
- We now refer to the DTT test as a test of speech-in-noise recognition or reception, in agreement with Smits et al (2013) JASA
- We also changed the labels in Figure 3 to avoid the term “noise tolerance”.

Furthermore, we agree with the reviewer that measures of hearing sensitivity (e.g., online audiometry) would have been desirable. However, online audiometric tests need careful validation to make sure that they do measure what they are supposed to measure. We are currently planning a follow-up study that will include such measures. In the revised Methods section, we state why no measures of hearing sensitivity were included in the present study (p. 17):

Of note, no additional direct measures of hearing sensitivity (e.g., mobile-based audiometry; Saliba et al., 2017) were included in the present online study in order to keep the overall study duration short. Furthermore, online audiometric tests would have required additional validation procedures, such as comparison of online- with lab-audiometry for a representative sub-sample of subjects.

My concern is that without accounting for hearing sensitivity in a way other than self-report, which is not highly correlated with behaviorally assessed hearing, it is not accurate to describe the results as showing a dissociation between subjective v. objective noise tolerance, but rather of different associations of personality with self-reported noise tolerance vs. speech perception in noise or hearing sensitivity.

That the DTT is capturing signal audibility and hearing loss can be seen in this data set, as the greatest effects on performance were the speaker/headphone the participant used (those who used headphones performed better than those who completed the task in the sound field (internal or external speakers)) and how old they were (older individuals performed much worse and, importantly, are also much more likely to have hearing loss). Neuroticism score, by comparison, had only a very small effect.

When viewed this way, it is not surprising that there is a difference between subjective reports of noise tolerance, and someone’s ability to recall speech in background noise or their overall hearing ability, and that these reports and abilities show different relationships with personality traits. Clarifying the language seems especially important given the small effects reported in the paper. An alternative interpretation could be that neuroticism and extraversion predict how well someone feels they do in noise or can deal with noise (which has already been established in the literature), but that overall hearing ability or ability to understand words in background noise have a less well-defined relationship with personality.

Authors: We agree with the reviewer that given the lack of a hearing sensitivity test in the current dataset, it is not possible to unambiguously capture whether performance differences in the DTT indicate differences in hearing sensitivity or coping with noise. We believe that the reviewer's concern has in part been resolved by the fact that we avoid the misleading term "noise tolerance" when referring to the DTT test in the revised manuscript. Instead, we use the term "hearing in noise", such that we now claim, in direct agreement with what the reviewer suggested, that our results suggest that personality associates with self-reported versus objectively assessed hearing in noise. In the revised Discussion, we moreover emphasize the implications of the lack of a hearing sensitivity test for the interpretation of results (p. 12):

In the present study, higher objective hearing in noise was to large extents explainable by younger age and use of headphones (versus tabletop or in-built loudspeakers). Opposite to our hypotheses, neuroticism exhibited a small positive association with objective hearing in noise under statistical control for demographic information and sound presentation as well as other personality traits: If a person sampled at random scored above average on neuroticism, the probability of their objective hearing in noise being also above average is about 52%. It is obvious that the association of DTT scores with neuroticism was considerably smaller than their association with age, which is known to covary with hearing thresholds. Since hearing sensitivity (i.e., audiometry) was not included in the present study, we cannot differentiate whether higher neuroticism relates to lower (i.e., better) hearing thresholds or to better hearing in noise, which are both captured by the DTT.

Furthermore, in line with the reviewer's suggestion, we added a paragraph to the Discussion. It emphasizes that the relation of personality and subjective measures was considerably larger than the relation with the DTT. Also, we directly adopted the reviewer's wording of the alternative interpretation (p. 15):

It is important to emphasize that the relation of personality with subjective hearing in noise was considerably larger than the association with objective hearing in noise. In this sense, a straightforward interpretation of the present study is that personality traits neuroticism and extraversion predict how well listeners feel they do in noise or how well they can deal with noise. In contrast, listeners' objective hearing sensitivity or hearing in noise show a smaller, less well-defined association with personality.

In lieu of rerunning the study and including an additional hearing screening measure, to allow the authors to parse out the role of hearing sensitivity on the self-report measures and DTT performance, the authors could consider reframing the study. One possibility is to replace the term "objective measure of noise tolerance" with the standard language used for the test, speech perception in noise. The authors may consider using the language they used for their OSF folder, assessing how personality traits predict noise sensitivity and speech in noise comprehension—though the DTT is not necessarily a measure of comprehension, as previously discussed. This is a more conservative, and in this reviewer's opinion, more precise description of what was assessed under the present study conditions.

Authors: In line with some of the previous concerns, we reframed the present article to avoid the term "noise tolerance". Also, the title of the article has been changed accordingly to:
Personality captures dissociations of subjective versus objective hearing in noise

In the revised manuscript, this concern has furthermore been addressed as we refer to the DTT test as a measure of speech-in noise reception / comprehension (in agreement with Smits et al (2013) JASA) and use the term "noise tolerance" only to refer to the WNSS and ANL tests. We hope that the reviewer agrees that framing the present results in the context of dissociations in "subjective versus objective hearing in noise" more adequately captures the present results.

As stated above, we are currently planning a large-scale follow-up study. The reviewer's comment encouraged us to include online audiometric measures (plus a validation of these measures) in the follow-up study.

Additional points:

Table 1 seems to suggest that several other personality traits contribute to self-reported noise tolerance on a similar scale as neuroticism and extraversion. Can the authors comment on why they did not include some thoughts on these results in their discussion? It could add to the literature to more fully characterize the contribution of personality to self-reported noise tolerance.

Authors: We agree with the reviewer that these additional associations of personality traits with hearing in noise tests deserve mentioning, although they were not of primary interest with respect to our hypotheses. At the end of the Introduction, we mention that these other BIG-5 personality dimensions were not directly related to our hypotheses (p. 4):

We focused particularly on these two personality dimensions since both have been consistently related to subjective hearing in noise in the literature (but also included remaining BIG-5 dimensions agreeableness, openness, and conscientiousness).

In the end of the Results section, we refer to these effects as follows (p. 10):

Beyond personality traits neuroticism and extraversion, Table 1 shows that other BIG-5 personality traits explained variance of hearing in noise tests. Higher self-reported noise resistance in the WNSS was associated with higher agreeableness ($\beta = .116$; $p < .001$; $CL = 0.538$) but with lower openness ($\beta = -.07$; $p = .021$; $CL = 0.477$) and conscientiousness scores ($\beta = -.091$; $p = .004$; $CL = 0.471$). Furthermore, better self-reported speech-hearing ability in the SSQ was associated with higher openness ($\beta = .101$; $p < .001$; $CL = 0.533$) and conscientiousness scores ($\beta = .122$; $p < .001$; $CL = 0.537$).

Since these results were not related to our initial hypotheses, we do not further discuss them here. As our data are available online, we would appreciate a lot if future studies use these data for further analyses.

Listening effort: The authors state in the methods that they asked questions about listening effort in the online study and they also bring listening effort into their discussion section, however they do not report on those findings in the present paper. Can the authors comment on why they did not include this variable in the current manuscript? This could be important, if those results support their point that the allocation of more cognitive resources for those with higher neuroticism could be one way to explain their findings of the dissociation between objective and subjective noise tolerance.

Authors: We agree with the reviewer and now briefly refer to these data in the Discussion section. Reviewer #2 raised a similar concern. Since we included only two questions (one asking for mental

demand and the other for expended effort in the DTT), we cannot formally evaluate reliability of these measures and we are thus careful to not over-interpret the results. The following section has been added to the revised Discussion (p. 14):

In the second cohort of the present study, we included two questions following the DTT to probe participants' experienced mental demand and expended effort. Under statistical control for potential confounders (age, gender, sound presentation, highest education, hearing loss), neuroticism was a significant predictor of expended effort ($\beta = 0.099$; $p = 0.031$) but not of experienced mental demand ($\beta = 0.077$; $p = 0.091$). These results are consistent with the view of higher expended listening effort with higher levels of neuroticism.

In the revised Methods section, we briefly introduce the two effort questions used in cohort 2 (p. 19):

After the DTT test, participants in cohort two completed two short questionnaires to assess some aspects related to listening effort. Two questions (adapted from the NASA-task load index; Hart, 2006) were intended to assess mental demand ("How high was your mental demand for this listening test?"; translated from German) and expended effort ("How hard did you have to work to reach your level of performance?"; translated from German) on a 10-point scale (1, very low; 10, very high).

The common language effect sizes are all very close to 50%, so while it is appreciated that they used CL effect sizes, it does little to assuage my feelings that the effects are very small, although the authors do not generally describe them as such. Can the authors please discuss the size of the different effects they identified and provide confidence intervals for these effects?

Authors: In the revised Discussion, we now emphasize in several instances that the effect sizes are actually very small (especially for the association of neuroticism with performance in the DTT). Furthermore, we compare effect sizes and state explicitly that the effects for the association of personality with subjective measures was considerably larger than the association of personality with objective performance in the DTT. We also emphasize the difference in the sizes of these effects in the Results section of the revised manuscript (p. 9&10):

The normalized difference extraversion_{z-score} – neuroticism_{z-score} explained inter-individual differences in subjective hearing in noise (WNSS: $\beta = 0.326$; $p < 0.001$, CL = 0.604; SSQ: $\beta = 0.193$; $p < 0.001$; CL = 0.562; ANL: $\beta = 0.166$; $p < 0.001$; CL = 0.552). This effect corresponds to a considerable displacement along the vertical axis in Figure 3 d. Although smaller in effect size, the extraversion–neuroticism difference score also explained objective hearing in noise (DTT: $\beta = -0.077$; $p = 0.0124$; CL = 0.475), corresponding to a small displacement of participants along the horizontal axis in Figure 3 d.

If the reviewers allow us one comment on their concern of small effect sizes in our study: While we agree that the reported key findings on personality are factually small, it deserves emphasis that, first, larger, more representative samples (like ours) almost inevitably yield smaller effect size estimates, irrespective of research area: "Small effect sizes from large-N studies are the most likely to reflect the true state of nature." (Funder & Ozer, 2019, p. 164). The small-N laboratory studies with correlations in the $r = .5$ range are themselves likely biased overestimates of true effect sizes (e.g., Albers & Lakens 2018).

Second, as we argue in the Discussion section, also with reference to Funder & Ozer (2019), small effects per se are not to be dismissed too quickly: If these effects occur repeatedly with relatively high

frequency in everyday life —and we think this is the case with hearing and all acts of communication—, then even minute effects sizes can have a considerable cumulative impact on an individual's behavioural outcomes, the behavioural choices they are taking, and so forth.

We also mention in the revised Methods section that confidence intervals for all effects (standardized coefficients) from regression analyses can be found in the online data.

Appendix B

Department of Psychology, University of Lübeck, Maria-Goeppert Straße 9a, 23562 Lübeck, Germany

Lübeck, October 07, 2021

Dear Royal Society Open Science editorial staff,

we are grateful that our article RSOS-210881 has been accepted for publication subject to minor revision.

We addressed all remaining minor comments raised by reviewer #3 (see point-by-point reply below).

Sincerely,

Malte Wöstmann and Jonas Obleser

Reviewer: 3

Comments to the Author(s)

RSOS-210881.R1

Personality captures dissociations of subjective versus objective hearing in noise
Wöstmann, Erb, Kreitewolf, Obleser

I thank the authors for their careful consideration of my comments on the initial submission. It is clear that they took all reviewer comments and concerns under advisement and made a considerable number of modifications, which have strengthened the manuscript. My major concerns have been adequately addressed. Below are some minor concerns for the authors to consider.

- P17 Ln 27: This reference is quite old and is likely not reflective of the current relationship between noise sensitivity and hearing aid use, given the considerable advances in hearing aids in the last 30 years (e.g. digital vs. analog systems, noise reduction algorithms, directional microphones, etc). The authors may consider removing this statement unless a more current paper supports this sentiment.

Authors: In agreement with the reviewer's suggestion, this statement has been removed.

- P18 Ln 8-23: This paragraph is a bit hard to follow. I recommend adding "of hearing loss" to the sentence on line 13, which will help that sentence and the one following it. Additionally, as written, line 17 implies that hearing aids have limited benefit for speech perception in noise. This is true for some hearing aid users, and likely affects their success/satisfaction with the devices but hearing aids DO generally improve speech perception in background noise. I believe the authors are trying to describe that for individuals who report limited hearing aid benefit in noise, audiologic and demographic factors do not fully describe this lack of benefit. Please consider revising, for clarity.

Authors: We revised this paragraph in line with the reviewer's suggestions.

- The authors may consider including the results of -Nabelek, Tampas, Burchfield (2004). Comparison of speech perception in background noise with acceptance of background noise in aided and unaided conditions. JSLHR. 47:1001-1011- in the background, as it directly relates to their goal of understanding subjective vs. objective hearing in noise.

Authors: In the revised manuscript, we cite this article.

- Throughout the document there are instances where the authors describe hearing in noise ability as high/low, which I felt detracted from the readability. It is likely that this is partially due to switching out "noise tolerance" with "hearing in noise" and in some cases it doesn't

quite work with the same wording. Below is a list of the instances that I found to be problematic/unclear and which the authors should consider modifying to include language to help the reader (e.g. worse/poorer instead of lower, better instead of higher)

P 17, Ln 57

P 19 Ln 10-11

P 23 Ln 3, 7

P 26, Ln 27

P 27, Ln 33

Authors: We are grateful that the reviewer made us aware of this. We changed the wording accordingly for all of these instances.